# TheMCPCompany: Creating General-purpose Agents with Task-specific Tools

## Abstract

Since the introduction of the Model Context Protocol (MCP), the number of available tools for Large Language Models (LLMs) has increased significantly. These task-specific tool sets offer an alternative to general-purpose tools such as web browsers, while being easier to develop and maintain than GUIs. However, current general-purpose agents predominantly rely on web browsers for interacting with the environment. Here, we introduce TheMCPCompany, a benchmark for evaluating tool-calling agents on tasks that involve interacting with various real-world services. We use the REST APIs of these services to create MCP servers, which include over 18,000 tools. We also provide manually annotated ground-truth tools for each task. In our experiments, we use the ground truth tools to show the potential of tool-calling agents for both improving performance and reducing costs assuming perfect tool retrieval. Next, we explore agent performance using tool retrieval to study the real-world practicality of tool-based agents. While all models with tool retrieval perform similarly or better than browser-based agents, smaller models cannot take full advantage of the available tools through retrieval. On the other hand, GPT-5's performance with tool retrieval is very close to its performance with ground-truth tools. Overall, our work shows that the most advanced reasoning models are effective at discovering tools in simpler environments, but seriously struggle with navigating complex enterprise environments. TheMCPCompany reveals that navigating tens of thousands of tools and combining them in non-trivial ways to solve complex problems is still a challenging task for current models and requires both better reasoning and better retrieval models.[1]

## 1 Introduction

Since the introduction of the MCP protocol by Anthropic in November 2024 (Anthropic, 2024; FastMCP, 2025), there has been continuous explosive growth in the number of MCP servers. A June 2025 survey by Virustotal (Quintero, 2025) counted 17845 MCP server projects on GitHub. The awesome-mcp-servers list (Gizdov, 2025) contains over 7000 publicly available MCP servers. And this is just public servers; more and more, organizations are creating MCP servers to expose the functionality of internal tools to LLMs as well. This makes sense for a number of reasons. Using MCP servers, LLMs can directly call the specific tools needed for completing each task (e.g., create_pr and merge_pr) (Patil et al., 2023; Schick et al., 2023). MCP servers are relatively simple to create and maintain, and providing direct access to tool documentation provides a straightforward way for LLMs to interact with new environments.

Despite this proliferation of direct access to tools and API surfaces, however, general-purpose agents still predominantly rely on general-purpose tools such as web browsers and code interpreters to solve problems (Fourney et al., 2024). Here, we aim to understand the capabilities and performance of an alternative approach: general-purpose agents based on large, heterogeneous tool collections.

Although there are several prior papers studying specific aspects of tool-based agents, none of them provides a comprehensive view of the challenges that come with the combination of a large number of tools and complex tasks in a complex environment. First, there is a growing body of work on creating general-purpose AI agents. While these works represent the complexity of tasks and

---

[1]We will release all our code and data after the double-blind review process.

environments that agents face in reality, they often incorporate a very small number of task-specific tools (e.g., a dedicated search tool) (Mozannar et al., 2025; Soni et al., 2025). Thus, it is unclear how AI agents behave when the number of available tools increases significantly. On the other hand, there is a rich literature that studies different challenges of tool calling with LLMs (Qu et al., 2025), such as complex function calls (Zhong et al., 2025) and large tool sets (Qin et al., 2023). However, tool calling works often rely on simple environments that are not representative of practical applications, like automating enterprise workflows. Our goal is to provide a realistic environment that includes challenging tasks, complex services, and a large and complex tool set for studying the potential and challenges of tool-based agents in practical scenarios.

We introduce TheMCPCompany, an extension of TheAgentCompany (Xu et al., 2024a) that simulates a software company where MCP tools are available for all operations in the company. In fact, this simulation represents our vision for enterprise environments in the future. To better represent complex enterprise workflows, we expand TheAgentCompany's environment by introducing the Microsoft Azure cloud computing platform[2]. We then create a fully functional MCP server for each of the services (Azure, Plane, GitLab, ownCloud, and RocketChat) that exposes its full functionality through tools (more than 18,000 tools in total, of which almost 17,000 come from Azure). We adapt the existing tasks from TheAgentCompany to the MCP setting and create a new set of tasks specifically for Azure. These tasks range from relatively simple ones whose solutions can be found in a web search to complex, enterprise-level debugging (Fig. 1). Finally, we annotate a small set of required tools for each task, allowing us to evaluate tool use separately from tool selection.

We also create MCPAgent, a baseline agent that treats tool retrieval itself as a tool. MCPAgent has access to all 18k tools, but it must discover them by constructing queries and then reasoning about the results. This allows the agent to explore different solution trajectories and dynamically search for the required tools and their dependencies. We implement MCPAgent based on OpenHands' CodeAct agent (Wang et al., 2024c).

We evaluate six different LLMs on the tasks adapted from TheAgentCompany and show that task-specific tools are a practical and even preferred interface for interacting with the environment. Compared to OpenHands' CodeAct agent, which uses a text-based browser, an agent with access to the ground truth tools improves performance by 13.79 points and reduces costs by \$2.29 per task on average (54% reduction in costs). Even without the ground truth tools, our MCPAgent with the tool-finder function outperforms the alternative browser-based agent by 5.39 points and reduces costs by \$2.06 per task on average. On these tasks, GPT-5 performs almost as well with the tool finder as with ground-truth tools.

In contrast, on our hardest tasks in the Azure environment, even the most capable reasoning models fail almost completely. We find that agents mainly struggle with the diversity and complexity of Azure services. For example, they fail to correctly identify the issue with a broken application, do not consider all possible solutions when one fails, and often implement only part of the solution.

Our results show that agents can solve problems in enterprise environments that are more complex and contain far more tools than previously considered in the literature. They also show that MCP is a key facilitator: exposing tools to LLMs via a standardized protocol leads to better results than relying on browser-based agents. However, our results also reveal a key challenge going forward in this space: navigating thousands or more tools that must be combined in non-obvious ways to solve complex problems is both a retrieval and a reasoning problem. The most advanced reasoning models are capable of searching for tools, but more work is needed on both fronts to fully realize our vision for future enterprise environments. TheMCPCompany supports this work by inviting future contributions to explore more realistic and complex scenarios that agents face in practice.

## 2 RELATED WORK

**AI Agents**   There is a growing body of work on AI agents (Handa et al., 2025; Shao et al., 2024; 2025; Wu et al., 2023; Xie et al., 2024). Although most of the first generation of agents are domain-specific, such as coding (Wang et al., 2024c; Xia et al., 2024; Yang et al., 2024) or browsing agents (Chezelles et al., 2024), more recently there has been a push toward general-purpose agents that can complete diverse tasks across multiple domains (Hu et al., 2025). Since a general-purpose

---

[2]https://azure.microsoft.com

agent needs to interact with different services depending on the given task, current agent frameworks predominantly interact with the environment via general-purpose tools such as a browser, shell, or Python interpreter (Soni et al., 2025). Recently, Song et al. (2024) proposed using REST API calls instead of browser interactions. However, compared to REST APIs, MCP tools are easier to create and are being actively developed by the machine learning community, and thus better suited for use with LLMs. Moreover, Song et al. (2024) use a small number of tools (less than a thousand) for each task and provide a short description of all tools in the prompt, which does not scale to large tool sets capable of performing in practical scenarios. For these cases, retrieval is necessary.

Agent benchmarks have also evolved in different directions. For example, there are many benchmarks that aim to create complex tasks (Mialon et al., 2023), simulate realistic environments (Xu et al., 2024a), or study the impact of agents on the workforce (Styles et al., 2024). However, similar to agent frameworks, these benchmarks are either limited to a small set of tools (Barres et al., 2025; Wang et al., 2024a; Yao et al., 2024) or mainly rely on the browser (Zhou et al., 2023) for agent interactions.

As a result, the challenges and opportunities for agents that primarily rely on large tool sets to interact with the environment are largely unknown. Here, we build on prior work (Xu et al., 2024a) and maintain the complexity and realism of the tasks and environment. However, we replace the few general-purpose tools with a large number of task-specific tools and investigate the challenges and opportunities that agents face in this new setup.

**Tool Use**  The ability to call tools to interact with the environment is what makes the current generation of AI agents feasible. There is an extensive body of research studying various aspects of tool calling with LLMs (Chen et al., 2025; Qu et al., 2025; Yuan et al., 2023), ranging from the complexity of tool calls (Zhong et al., 2025) to dependency between tools (Lumer et al., 2025). However, most works rely on a small set of tools and do not represent the growing scale of MCP tools available to LLMs (Dong et al., 2025; Feng et al., 2025; Li et al., 2025; Wang et al., 2025). While there are several works that investigate large tool sets, their environments are simple compared to what agent benchmarks provide (Fei et al., 2025; Gan & Sun, 2025; Liu et al., 2024a; Qin et al., 2023; Shi et al., 2025; Xu et al., 2024b). The tasks are also simple and often there is significant semantic overlap between the task description and tool specifications, which simplifies tool selection (Li et al., 2023; Liu et al., 2024b). However, in practice, task descriptions (e.g., fix a broken app) often do not mirror the name and description of the required tools (e.g., list_managed_identities).

With the increasing popularity of MCP, there is a renewed interest in tool calling benchmarks but through MCP servers (Lei et al., 2025; Luo et al., 2025b). What sets MCP tools apart from traditional tool calling is the opportunity for massively scaling the number of tools by standardizing the communication protocol. However, current MCP benchmarks are generally limited to between a few hundred and a thousand tools (Gao et al., 2025; Liu et al., 2025; Luo et al., 2025a; Mo et al., 2025; Yin et al., 2025). Moreover, the related MCP servers for each task are manually selected for the agent prior to execution which ignores the impact of tool selection as one of the main challenges that agents face when dealing with large tool sets (Luo et al., 2025b).

Unlike prior work on tool calling, we take full advantage of MCP's main strength, scalability, and create more than 18,000 functional tools for interacting with different real-world services. Also, in our setup, we do not directly provide the related tools for each task to the agent. Instead, it needs to use a tool finder function to search for and discover the required tools on its own.

## 3 THE MCP COMPANY

Considering the simplicity of developing and maintaining MCP servers and the growing interest of the community, we argue that in the near future, MCP tools will be LLMs' primary interface for interacting with the world. In other words, there will be an MCP tool for every operation and every application (e.g., GitLab); teams in an organization will also offer MCP servers for interacting with their services. In fact, this is already happening. Many services already offer MCP servers, and there are numerous efforts to further simplify widespread adoption of MCP. For example, Docker

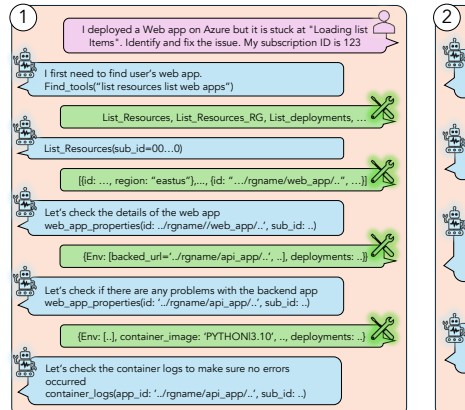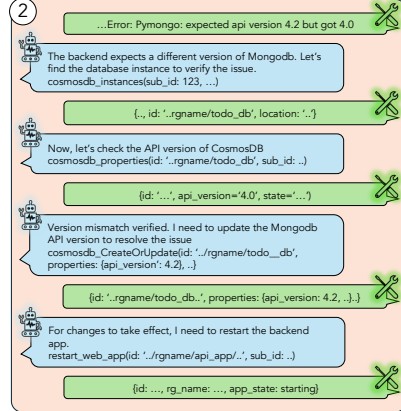

Figure 1: The correct solution trajectory for one of our complex Azure tasks. **Note:** the agent must use the tool finder function to discover each one of the tools used in the trajectory. But, due to space constraints, here, we only show the first call to `find_tools`.

Desktop offers a dedicated toolkit that simplifies the deployment of MCP servers (Docker, 2025), and there is even a registry for keeping track of the growing number of MCP servers[3].

Here, we describe TheMCPCompany, an extension of TheAgentCompany benchmark that simulates a realistic and tool-rich environment. We first include the Microsoft Azure cloud computing platform, which significantly increases the complexity of the environment and the number of available actions. Then, we create MCP servers for all services in TheMCPCompany that expose the full functionality of each service through a large collection of tools.

## 3.1 THEAGENTCOMPANY (BACKGROUND)

TheAgentCompany is a benchmark that simulates a small-scale software company for evaluating agents' ability to complete everyday enterprise tasks (Xu et al., 2024a). TheAgentCompany offers self-hosted services for project management (Plane), DevOps (GitLab), communication (RocketChat), and productivity (ownCloud) as docker images with pre-populated data. It also configures LLM-powered non-player characters that act as employees in the company. This provides a realistic environment, where the agent needs to interact with multiple services and simulated employees to successfully complete a task. For each task, TheAgentCompany provides an evaluation script, with multiple checkpoints, that assigns partial credit when the agent only completes part of the task.

We choose TheAgentCompany as the basis for our work since its environment approximates real-world applications more closely than other benchmarks. More importantly, the action space provided by the four hosted services is considerably larger than that of other agent benchmarks, which facilitates our goal of creating an environment with a large tool set for agent evaluation.

## 3.2 AZURE TASKS

While TheAgentCompany uses real-world applications for simulating the environment, these self-hosted applications are simpler than many services used in production by most organizations. For example, most software companies use cloud computing platforms, such as Azure and AWS, as part of their workflow. These services are so complex that employees take dedicated courses just to be able to manage the infrastructure. Therefore, to have a realistic view of LLMs' potential for practical applications, we require an environment where agents directly work with services used in production, instead of simpler proxies commonly used for evaluation. To achieve this, we have created two small sets of tasks that require managing resources in the Microsoft Azure cloud platform. Our tasks exercise a range of activities that require interacting with different parts of Azure, including resource management, security, storage, compute and Cognitive Services like image recognition.

---

[3]gh/modelcontextprotocol/registry

In the first category, we have created 10 *primitive* tasks, where the agent only needs to take a very specific action on a very specific resource. Examples of primitive tasks are adding tags to a given resource or deleting a specific resource. These tasks mainly measure agents' ability to identify the correct tool for a given action from the large pool of Azure MCP tools and generate the correct tool call. For the second category, we have created seven *composite* tasks that are intended to reproduce more challenging real-world scenarios that an Azure user would normally have to carry out (Fig. 1). The composite tasks involve an infrastructure with multiple services (e.g., CosmosDB, Key vault, Function app) that are configured for a specific application, like serving a TODO list web app. In this category, the agent is given higher-level goals, such as fixing a broken app, implementing a security policy, or adding a new feature. To successfully complete the composite tasks, the final state of the environment must meet the requirements of the task in addition to having a working application. The composite tasks are more difficult and measure the agent's ability to understand and navigate the complex logic of the Azure environment, such as coordinating code edits and environment configuration and understanding the space of possible solutions for a given problem.

**Task Details**  To make evaluations more accessible, our tasks use the cheapest Azure resources and can be run using a free-tier Azure subscription (free Azure subscriptions come with a $200 credit. During the development and troubleshooting of the tasks, which involved executing each task many times, we spent less than $1 of this limit). For each task, we provide a task description, an evaluation script to judge whether the task was completed successfully, and a proof-of-concept script that solves the task using the available MCP tools. Moreover, to have a reproducible environment, we provide a Terraform[4] script for each task that initializes and tears down the execution environment on Azure.

## 3.3 A Large and Comprehensive Tool Set

To provide an environment where the agent primarily relies on task-specific tools for interacting with different services, we create a large collection of tools that collectively expose the full functionality of each of the services in the environment. For example, we create dedicated tools for merging a PR on GitLab or listing the resources in an Azure subscription.

Most modern services come with comprehensive REST APIs that offer a dedicated endpoint for each operation (e.g. list available users). While prior work has proposed agents that directly call the REST APIs (Song et al., 2024), we argue that MCP tools are a more appropriate solution for large-scale adoption in long term. Thanks to libraries like FastMCP (FastMCP, 2025), MCP tools are easier to develop and maintain compared to REST APIs. More importantly, MCP tools are LLM friendly: each tool is accompanied by the description of its functionality and arguments, and MCP provides an easy and standard method for accessing these documentations. This allows LLMs to discover the required tools for each task and also learn how to use new tools on the fly. On the other hand, there is no standard method for providing the REST API documentations to LLMs. Therefore, we convert the REST APIs of Azure, GitLab, and RocketChat into dedicated MCP servers that provide a corresponding tool for each API endpoint. We also extract the description for each tool and its arguments from the API specifications provided by each service. See Appendix F for details.

Plane and ownCloud do not provide comprehensive REST API support. To overcome this, we treat own-Cloud as a file server and manually create an MCP server that provides basic file operations (e.g., download and upload). We observe that these file operations are sufficient for completing TheAgentCompany tasks, and the agent often uses Python libraries to manipulate the spreadsheet or presentation files on ownCloud. Finally, we adopt the official MCP server for Plane and manually add any missing tools that are required for completing the tasks. After creating the MCP servers, we manually go through all the tasks and make sure they are feasible with the available tools.

| Service | #MCP Tools | Avg #Args | Complex Tools (%) |
|---|---|---|---|
| Plane | 52 | 2.06 | 28.85 |
| RocketChat | 520 | 2.82 | 12.31 |
| ownCloud | 11 | 1.64 | 0.00 |
| GitLab | 1,085 | 5.47 | 10.69 |
| Azure | 16,837 | 5.63 | 22.50 |
| Total | 18,505 | 5.53 | 21.52 |

Table 1: The number and properties of tools provided by TheMCPCompany.

---

[4]hashicorp/terraform

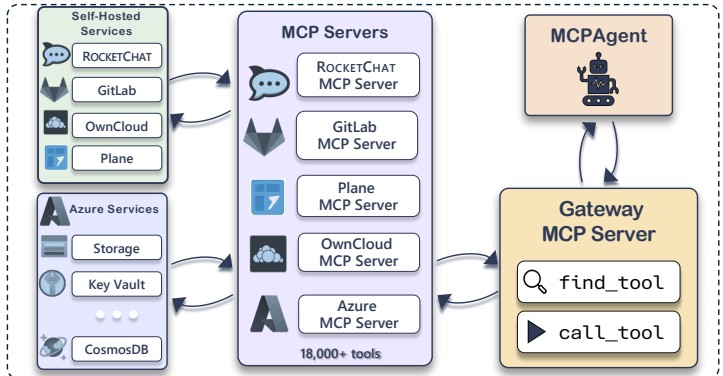

Figure 2: Our MCP servers expose the full functionality of each service through tools. Instead of directly providing the 18,000+ tools to the agent, we provide it with a gateway MCP server with two tools, which the agent can use to search for and invoke the required tools at each step.

**Tool Characteristics** In addition to providing a large number of tools, TheMCPCompany's tool set also represents the complexity of tool calls in practice (Table 1). On average, our tools accept more than five arguments and, in some cases, the agent has to provide up to 39 arguments for some tool invocations for Azure. For example, to create a virtual machine, the agent should provide detailed information about all the dependent resources (e.g., disk, network interface, virtual networks, OS image, role assignments, etc.). There is also a significant dependency between our tools. For instance, the agent has to first create all the dependent resources in order to be able to successfully call the tool for creating a virtual machine. Moreover, many of TheMCPCompany's tools require passing arguments with complex data types. Specially, for Azure and Plane, 22.5% and 28.85% of tools have at least one argument of type array or object. For Azure, this is more than 3K complex functions, and in our experience, most of the tools that change the environment state (e.g., create or modify resources) require deeply nested arguments.

Moreover, our tool set represents the chaotic nature of real-world applications. For example, there are similar tools with totally different purposes (e.g., send_msg_to_room, send_msg_to_individual). On the opposite side, often there are several tools for each action, with slight differences (e.g., gitlab_search_all, gitlab_search_issues). Similarly, there are different sequences of tool calls for accomplishing a goal, with some more efficient than others.

**Task Modifications** We update the task descriptions and evaluation scripts in TheAgentCompany, which are written for browser-use agents, to be compatible with tool-based agents. Furthermore, for each task, we annotate a small set of tools that are sufficient for its successful completion. Later, we use these annotated tools to isolate the impact of tool selection and measure the upper bound on the performance of tool-based agents with current models. See Appendix D for more details.

## 4 MCPAGENT

We create a baseline agent to study the feasibility of tool-based agents with a large tool set (Fig. 2). Utilizing the extremely large number of tools is the main challenge for creating practical tool-based agents. Naive solutions are untenable; the context window of current LLMs does not fit the specification for all the tools in our benchmark (18,000+). To address this issue, prior work uses retrieval models to select the necessary tools based on task descriptions (Qin et al., 2023). However, for realistic and challenging tasks, such as those in TheAgentCompany, the task description often has little in common semantically with the description of the required tools. For example, while role assignment is necessary for managing storage accounts in Azure, there is little semantic similarity between tools related to role assignment and the description of a task for backing up a storage account.

Instead of selecting the tools prior to execution, we allow the agent to select the tools itself. Specifically, we create a gateway MCP server with a tool finder function that the LLM can use to search for required tools at each step using a text query. Under the hood, the tool finder uses a text embedding model to encode the JSON specification of the tools and also the agent's query. Then, based on the cosine similarity between query and tool embeddings, it returns the specification for the top-k

tools (see Appendix E.1 for more details). Since the LLM does not have direct access to the main tools, the gateway MCP server provides another function that takes the name and arguments of any of the retrieved tools, calls the tool for the LLM, and returns the results. This architecture keeps the number of tools manageable for the agent, and at the same time, it provides more flexibility by allowing the LLM to explore different solutions and choose the required tools dynamically. Moreover, it also provides a unified interface to a heterogeneous set of tools. Finally, except for the browser tool, our agent has access to all the standard tools in OpenHands' CodeAct agent (Wang et al., 2024b;c) (Think, Python, Shell, Web fetch, and File edit), which are necessary for completing TheAgentCompany tasks.

## 5 EXPERIMENTS

### 5.1 SETUP

We build our agent based on OpenHands' CodeAct agent (Wang et al., 2024c), with a slightly modified system prompt that instructs the LLM to use tools instead of the browser. See Appendix E for details. We then evaluate GPT-4.1, o3, GPT-5-mini, GPT-5, Sonnet-4, and Opus-4.1 on TheAgent-Company and Azure tasks (Anthropic, 2025; OpenAI, 2025). We use OpenAI's text-embedding-3-large model to calculate the embeddings for the tool finder function (OpenAI, 2025). Unfortunately, because of incompatibility with OpenHands, we disable the thinking blocks for Opus-4.1.

**Evaluation** For TheAgentCompany tasks, we use the same evaluation metrics as Xu et al. (2024a). The score for each task consists of two parts. The obtained credit from evaluation checkpoints accounts for 50% of the final score. The other 50% is only assigned if the agent completes the task successfully. We also report the percentage of tasks completed successfully and the average steps and inference costs for each task. The inference costs are calculated based on the token usage for each task and prices published by LLM providers. Since there are many valid solution trajectories for Azure tasks, we only consider the successful completion for evaluation without partial credits.

### 5.2 THEAGENTCOMPANY TASKS

**Potential of Task-specific Tools** First, we consider the question of whether task-specific tools are an appropriate interface for interacting with the environment. We directly provide the small oracle tool set to the agent for each task, excluding the impact of tool retrieval on performance. Compared to OpenHands' default CodeAct agent, which uses a text-based browser, using task-specific tools increases performance by 13.79 points on average across different models, with more than 20 points for o3 (columns Browser and Oracle Tool Set in Table 2). Except for GPT-5 which has good performance in both cases, we observe that the reasoning models, Opus-4.1 and o3, benefit more from task-specific tools than do their non-reasoning counterparts (Sonnet4 and GPT-4.1).

While with a browser, the agent needs to navigate the web interface and process the entire content of each web page, task-specific tools allow the agent to take the necessary action directly and only process the required information, which reduces inference costs. Across different models, the agent with the oracle tool set reduces inference costs by $2.29 on average per task compared to the browser-based agent, with up to $7.41 reduction in average costs per task for Opus-4.1. Moreover, for all models except for Opus-4.1 and o3, the number of required steps for each task also decreases, which directly translates to latency and usability of the resulting agents. The combination of better performance and reduced costs positions large sets of task-specific tools as a promising approach for developing general-purpose agents.

**Task-specific Tools in Practice** In real-world applications, we do not have access to the oracle tool set. To investigate the feasibility of creating general-purpose agents with task-specific tools in practice, we evaluate MCPAgent, which uses tool retrieval to discover the necessary tools for each task (Table 2). We find that even without the oracle tool set, using task-specific tools is preferred over the browser. Compared to the browser-based agent, MCPAgent improves performance by 5.39 points on average across all models, with a maximum improvement of 14.86 points for o3. Interestingly, the increases in performance are consistently larger for reasoning models compared to their non-reasoning counterparts.

| Model | Browser | | | | MCPAgent | | | | Oracle Tool Set | | | |
|---|---|---|---|---|---|---|---|---|---|---|---|---|
| | Score | Success (%) | Steps | Cost ($) | Score | Success (%) | Steps | Cost ($) | Score | Success (%) | Steps | Cost ($) |
| Sonnet 4 | 45.06 | 34.86 | 31.16 | 5.02 | 48.79 | 39.43 | 30.82 | 2.75 | 56.36 | 47.43 | 26.97 | 2.13 |
| Opus 4.1 | 41.16 | 31.43 | 24.07 | 14.58 | 48.68 | 39.43 | 22.53 | 7.29 | 57.26 | 48.00 | 23.65 | 7.17 |
| GPT 4.1 | 31.71 | 22.99 | 22.71 | 1.72 | 37.10 | 27.43 | 20.48 | 0.75 | 46.76 | 36.00 | 16.05 | 0.56 |
| o3 | 30.53 | 22.86 | 21.92 | 1.17 | 45.39 | 37.14 | 23.41 | 0.83 | 50.63 | 40.57 | 22.53 | 0.65 |
| GPT-5-mini | 33.36 | 24.57 | 31.74 | 0.41 | 32.11 | 22.86 | 29.27 | 0.26 | 49.33 | 38.86 | 22.33 | 0.17 |
| GPT 5 | 50.24 | 40.00 | 28.75 | 2.20 | 52.32 | 42.29 | 19.39 | 0.85 | 54.45 | 44.57 | 17.54 | 0.66 |

Table 2: The performance of different LLMs on the 175 tasks adapted from TheAgentCompany. Browser: the LLM uses the browser for completing tasks. MCPAgent: the LLM uses the tool finder function to discover and invoke the required tools. Oracle Tool Set: the LLM is provided with the required tools for each task.

Without the oracle tool set, LLMs cannot take full advantage of task-specific tools, and their performance is, on average, 8.4 points behind the agent with access to ground truth tools. We believe this gap would decrease in the future as the capabilities of LLMs improve. In fact, GPT-5 already closes the gap, and its performance without the oracle tool set only decreases by 2.13 points. However, this is the exact opposite for smaller and more affordable models like GPT-5-mini. In fact, the performance of GPT-5-mini without the oracle tool set is worse than its performance with the browser tool.

Interestingly, despite the additional calls to the tool finder function, MCPAgent provides similar cost savings to the agent with access to oracle tool set. Compared to OpenHands' CodeAct agent, MCPAgent reduces inference costs by $2.06 on average per task across all models. Our results show that even with current models, creating general-purpose agents with task-specific tools instead of a few general-purpose tools is practical and also provides significant benefits. These findings encourage future work to explore more effective agentic solutions for taking advantage of the growing number of task-specific tools available to LLMs.

| Model | Primitive | Composite |
|---|---|---|
| Sonnet 4 | 9/10 | 1/7 |
| Opus 4.1 | 9/10 | 1/7 |
| GPT 4.1 | 5/10 | 0/7 |
| o3 | 6/10 | 1/7 |
| GPT-5-mini | 2/10 | 0/7 |
| GPT 5 | 9/10 | 1/7 |

Table 3: The number of successfully completed Azure tasks in each category using MCPAgent with different LLMs.

## 5.3 Azure Tasks

Given the large action space of the Azure environment, we first use our primitive Azure tasks to evaluate if LLMs can correctly find and invoke the correct tool to achieve a very specific and clear goal, such as deleting a virtual machine (Table 3). We find that GPT-5, Sonnet-4, and Opus-4.1 use the tool finder function effectively and achieve nearly perfect scores on our primitive Azure tasks. However, GPT-4.1, o3, and GPT-5-mini struggle even with these simple tasks. Also, surprisingly, despite clear instructions to use MCP tools, GPT-4.1 and o3 often insist on using command line tools for interacting with Azure, and after they fail, they just provide a high-level outline of the solution and give up.

Evaluation on our composite tasks shows that LLMs' problem-solving capabilities diminish when faced with complex tasks in a complex environment, and all models consistently fail on almost all these tasks. We find that after failure, models do not explore alternative solutions. For instance, if the model does not have enough quota to deploy an Azure function, it does not try a different region or deploy the app on other resources like a container. Moreover, models do not follow a systematic approach for diagnosing and resolving problems. Instead, they focus on the most common cause for a given problem, often Identity and Access Management (IAM), and do not even check if their solution resulted in a functioning infrastructure.

## 5.4 Tool Calling Patterns

**TheAgentCompany Tasks** Table 4 reports the tool-use statistics of each model for TheAgent-Company tasks. LLMs effectively use the tool finder function and find the required tools after retrieving only about 20 tools, which is well below the maximum number of tools allowed by infer-

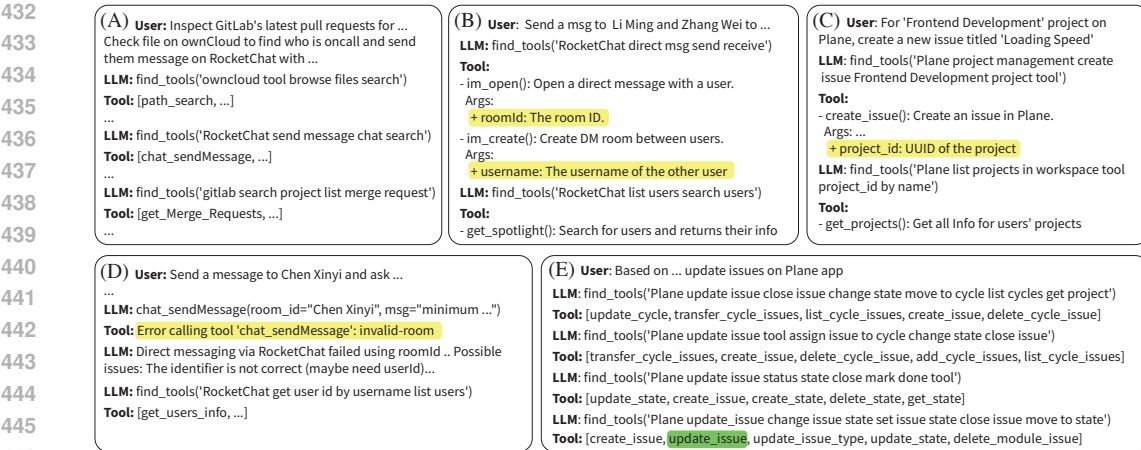

Figure 3: MCPAgent tool discovery patterns. A) Using a separate search query for each sub-task. B, C) Inferring tool dependencies from arguments of retrieved tools. D) Inferring tool dependencies from error messages. E) Persistently trying different queries to find the correct tools.

ence APIs (often 128). Also, solving each task requires only a handful of calls to task-specific tools, which explains the reduced inference costs of tool-based agents.

We find that reasoning models are better suited for use with a large number of task-specific tools. First, reasoning models call the MCP tools more accurately and fail less often than non-reasoning models. Similarly, reasoning models use tool retrieval more effectively and consistently achieve better retrieval recall. Finally, among the models that we tested, GPT-5 generates the most comprehensive and longest queries, which could explain its superior performance with MCPAgent.

**Azure Tasks**  Table 5 in the Appendix reports these statistics for Azure tasks, with similar patterns. One interesting observation is that the complexity of the tasks is also reflected in models' tool calling patterns. Except for GPT-4.1, o3, and GPT-5-mini that often fall back to command line tools and fail, other models consistently retrieve and call more tools for composite tasks than primitive tasks. Also, calling the correct tools with correct requirements and arguments is more challenging for composite tasks and consequently, the agent's tool calls fail more often. For composite tasks, identify-

| Model | #Retrieved Tools | #MCP Calls | Failed Calls (%) | Retrieval Recall | Query Length |
|---|---|---|---|---|---|
| Sonnet 4 | 15.7 | 9.9 | 10.7 | 60.0 | 34.5 |
| Opus 4.1 | 25.8 | 7.3 | 8.5 | 69.7 | 32.6 |
| GPT 4.1 | 13.5 | 9.1 | 29.7 | 44.9 | 31.6 |
| o3 | 22.2 | 7.8 | 13.0 | 53.1 | 19.2 |
| GPT-5-mini | 20.2 | 8.1 | 22.2 | 32.8 | 44.6 |
| GPT 5 | 15.3 | 11.5 | 8.3 | 58.7 | 52.9 |

Table 4: MCPAgent's tool calling statistics on the 175 modified tasks from TheAgentCompany. Query length is measured in number of characters.

ing a solution and retrieving the required tools is also difficult, and agents use the tool finder function more often and with longer queries.

## 6 ADDITIONAL ANALYSIS

**Retrieval**  To study the impact of better retrieval models on the performance of smaller models, we evaluate MCPAgent with GPT-4.1 and GPT-5-mini using the 0.6b and 4b versions of Qwen3-Embedding (Zhang et al., 2025) (Table 7 in Appendix C). Using better retrievers improves GPT-4.1's performance but does not have a meaningful impact on GPT-5-mini's performance. Considering that GPT-5-mini also failed to effectively use retrieval in our main experiments, these results further support our claim that better retrieval and reasoning models offer complementary benefits and neither is a replacement for the other.

We also compare MCPAgent's retrieval performance with standard retrieval, where we use the task descriptions as queries for tool retrieval (Table 8 in Appendix C). While standard retrieval only achieves a Recall@20 of 16.2, MCPAgent with GPT-4.1, which is the least effective among models

that benefit from retrieval, achieves a recall of 44.9 by retrieving only 13.5 tools on average. First, this illustrates that the description of complex tasks does not directly mirror the description of the required tools. Moreover, this demonstrates the effectiveness of MCPAgent in generating appropriate queries that find the required tools for each step of the task.

**Tool Discovery**    We provide several examples of MCPAgent's exploration patterns for finding the required tools for each task. First, LLMs break each task into several sub-tasks with specific goals (e.g., send a message) and generate a separate search query for each sub-task (Fig. 3A). Crucially, MCPAgent often discovers inter-tool dependencies from environment feedback. Stronger models, such as GPT-5, frequently infer tool dependencies based on the arguments of retrieved tools and issue new search queries accordingly. In Fig. 3B, the agent finds the tool for sending a text message, but immediately realizes that it requires the person's username. It then issues a new search query to find tools that can obtain the person's username based on their name. However, less capable models, like GPT-4.1, often do not proactively identify tool dependencies. Instead, errors from the environment trigger a search for additional tools (Fig. 3D). Finally, the persistence of reasoning models in trying different search queries or entirely different solutions after failure is a major contributor to success, especially for GPT-5 (Fig. 3E).

**MCPAgent Error Analysis**    To understand the challenges introduced by tool retrieval, we inspect trajectories where GPT-4.1 succeeds with the oracle tool set but fails with tool retrieval. We found that search errors accounted for a smaller proportion of task failures than one might expect. Instead, poor instruction following is a major cause of failure with two recurring patterns. In 33% of failures, the agent deviates from the task instructions when it finds seemingly relevant tools. For instance, if the task requires sending a direct message but the agent first finds a tool for sending a message to a channel, it does so and tags the user, which is not the correct solution. Surprisingly, for 50% of these tasks, the agent first finds the correct tools but then fails to follow the detailed task instructions. For example, it is asked to delete all GitLab repositories, but it only deletes some of them. Or it ignores the formatting requirements when creating a spreadsheet. We know the model is perfectly capable of using these tools to complete the task since the same model succeeds with the oracle tool set. We speculate that the increased cognitive load and context length from tool retrieval reduces the model's instruction following capabilities. See Appendix B for more details.

**Azure Composite Tasks**    On our most challenging Azure tasks, models' systematic problem-solving skills diminish significantly, with several recurring patterns. For example, LLMs assume the most common cause for a bug without verification (e.g., assuming secret management issues for database connection problems), implement only part of the solution (e.g., changing Azure's access settings without changing the application code), do not try other solutions if one fails (e.g., not trying different resource types or regions when out of quota), and often do not check if the implemented solution was successful (see Appendix A for more details and examples). This is in contrast to the models' behavior in the simpler TheAgentCompany environment, where they often systematically look for bugs, test their final solution, and explore other solutions if one fails. Our results encourage future work to further investigate the limits of LLMs' problem-solving skills in complex environments, which provides valuable insights for training better models in the future.

# 7 CONCLUSION

In this work, we introduce TheMCPCompany, a benchmark for general-purpose agents that primarily use task-specific tools for interacting with the environment. We provide MCP servers with a large number of tools (more than 18,000) that expose the full functionality of several real-world services. Our tool set is created from existing REST APIs and thus closely simulates tool calling in the real world. In addition, we include Microsoft Azure cloud computing platform in our environment and provide the necessary tools for all possible interactions with Azure, which significantly increases the environment's complexity. Through extensive experiments, we show the significant potential of task-specific tools for improving performance and reducing costs compared to browser-based agents. We also use tool retrieval to create a practical agent that automatically discovers the necessary tools for each task. We find that, even with imperfect retrieval, using task-specific tools still improves performance and reduces inference costs. Our results encourage future work to explore task-specific tools as an alternative approach for creating general-purpose agents. Also, the integration of Azure in our environment provides a valuable opportunity for future work to create more challenging tasks and further explore the agents' behavior in a real enterprise environment.

## LIMITATIONS

**Unintended Consequences of Deploying LLM Agents in Practice**  While providing the full functionality of production services, like Azure, to LLM agents opens a whole new category of tasks that LLMs can accomplish, it also increases the risks. Without any restrictions, deploying LLM agents in practice comes with many risks, such as destroying critical resources, incurring unnecessary costs (e.g., deploying expensive services), or exposing sensitive information to unauthorized users. For example, in our Azure tasks, GPT-5 mistakenly deletes a virtual machine, which is an irreversible action. While our work mainly focuses on the ability of agents to complete a given task, this is not sufficient for using LLM agents in practice. In addition to improving LLMs' performance, we encourage future work to also investigate potential approaches for mitigating the side effects of LLM actions without limiting the available actions to the LLM, for example, through human-in-the-loop agentic systems (Mozannar et al., 2025). By incorporating Azure, TheMCPCompany provides a realistic environment for future work to investigate different aspects of LLM agents in practical applications.

**Number of Azure Tasks**  Our Azure tasks reveal the weaknesses of LLM agents in navigating complex real-world environments. However, considering the numerous Azure services, there are many other types of problems and scenarios that are not included in our tasks. TheMCPCompany exposes the full functionality of Azure through tools. To better understand LLMs' behavior in enterprise workflows, we encourage future work to use TheMCPCompany's large tool set and investigate LLMs' behavior on other tasks and types of problems, such as multi-subscription governance, threat detection, and disaster recovery.

## ETHICS STATEMENT

Although the artifacts and methods presented in our work do not raise any immediate ethical concerns, incorporating LLM agents in actual production workflows requires extensive supervision and careful analysis, especially when interacting with user data. For example, in some of TheAgentComany tasks, the LLM is tasked to review several resumes and select the most qualified candidate. Delegating such tasks to LLM agents requires careful consideration since LLMs' biases could adversely impact parts of society (Bender et al., 2021).

## REPRODUCIBILITY STATEMENT

In our work, we use the same environment as TheAgentCompany (Xu et al., 2024a), which is based on publicly available docker images and creates the same container for all experiments. To create a reproducible environment for Azure tasks, we rely on the infrastructure-as-code paradigm. Specifically, we provide Terraform scripts for every task that create the same resources for each task every time and also destroy the resources at the end, to avoid extra costs. Moreover, we exclusively rely on the cheapest Azure services and the free credit assigned to all users, which ensures everyone can reproduce our results on Azure tasks. We use the default OpenHands (Wang et al., 2024c) parameters in our experiments and explain the exact version of OpenHands in our experiments as well as any modifications in Appendix E. Finally, to facilitate further progress in this direction, we will also release our data and code (including our MCP servers) to the public after the double-blind review process.

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

| Model | #Retrieved Tools | #MCP Calls | Failed Calls (%) | #Retrieval Attempts | Query Length |
|---|---|---|---|---|---|
| *Primitive* | | | | | |
| Sonnet 4 | 19.1 | 9.5 | 22.1 | 4.2 | 42.8 |
| Opus 4.1 | 33.0 | 8.2 | 8.5 | 3.7 | 39.1 |
| GPT 4.1 | 10.8 | 5.6 | 39.3 | 2.7 | 33.6 |
| o3 | 24.2 | 2.6 | 11.5 | 2.9 | 23.8 |
| GPT-5-mini | 15.4 | 2.8 | 25.0 | 0.9 | 44.9 |
| GPT 5 | 22.5 | 9.7 | 17.5 | 3.8 | 67.0 |
| *Composite* | | | | | |
| Sonnet 4 | 37.7 | 12.0 | 23.8 | 8.7 | 45.4 |
| Opus 4.1 | 59.0 | 10.7 | 16.0 | 6.9 | 44.3 |
| GPT 4.1 | 14.0 | 4.4 | 54.8 | 3.4 | 49.7 |
| o3 | 6.4 | 0.9 | 33.3 | 1.3 | 30.0 |
| GPT-5-mini | 15.8 | 1.3 | 11.1 | 1.1 | 92.0 |
| GPT 5 | 29.6 | 13.6 | 25.3 | 6.0 | 89.4 |

Table 5: MCPAgent's tool calling statistics on our primitive and composite Azure tasks. Query length is measured in number of characters.

## A    DETAILED ANALYSIS OF AZURE COMPOSITE TASKS

As mentioned in Section 6, the systematic problem-solving skills of even the best LLMs decrease significantly on our hardest Azure tasks. Here, we describe the setup and the solutions for three of our composite Azure tasks and then discuss the behavior of GPT-5 and Opus-4.1 for these tasks.

**Broken Web App CosmosDB Connection**    The Azure infrastructure for this task includes several resources (e.g., App Service and Key Vault) that are deployed for serving a simple TODO list web app. The main components are a front-end web app that sends requests to a backend web app, which then communicates with a CosmosDB MongoDB instance[5]. However, the MongoDB instance is configured to serve a different API version from what the backend web app expects, causing the web app to fail to load properly. To complete the task successfully, the agent should first troubleshoot the infrastructure and find the cause of the issue (i.e., API version mismatch) and then update the database configuration to fix the issue. Note that the task only describes the problem from the end user's perspective without discussing the infrastructure, i.e., my web app is stuck at "Loading list items", find the issue and fix it.

Below are the steps for one possible solution using available MCP tools:

- Use `Resources_ListByResourceGroup` to find the front-end and backend Web App instances.
- Inspect the Web Apps with `WebApps_GetConfiguration` to find their type (Linux containers in this case).
- Inspect the Linux container logs with `WebApps_GetWebSiteContainerLogs`, which show the exact error message about the database API version mismatch.
- Use `DatabaseAccounts_ListByResourceGroup` to find the resource ID of the target CosmosDB instance.
- Use `DatabaseAccounts_Get` to verify the current database API version.
- Use `DatabaseAccounts_Update` to update the database API version.
- Restart the web app with `WebApps_Restart` for changes to take effect.

---

[5]Adapted from `github.com/Azure-Samples/todo-python-mongo-terraform`

For this task, GPT-5 fails to find the correct tool to check the application logs, and thus, fails to identify the issue correctly. However, instead of further exploration, it assumes that access management issues and database secrets are the cause of the problem without any evidence. Interestingly, based on the thinking traces, the model itself is aware that there is no concrete evidence for this assumption. The following is part of the agent's thinking process: `App settings showed AZURE_COSMOS_CONNECTION_STRING_KEY referencing KV secret name, and AZURE_KEY_VAULT_ENDPOINT pointing to Key Vault. But Key Vault permissions` underline{`likely`} `not working, and app can't resolve secret`. After this point, GPT-5 keeps updating the secrets and different application settings until it runs out of budget and fails. See Appendix G.1 for the agent trajectory.

Opus-4.1 goes further and correctly identifies the source of the problem and generates the following as part of its reasoning process: `The API app is failing to start because of a PyMongo version incompatibility with Azure Cosmos DB`. Then, Opus-4.1 changes different configuration options for the web app, such as startup command, requirements, post-build command, etc. However, none of these changes resolves the issue, and the web app remains broken. See Appendix G.2 for the agent trajectory.

**Instant MongoDB to Blob Storage Backup**   This task uses the same infrastructure as the previous task, but without the bug. Instead, the model is asked to deploy a process on Azure that runs continuously and immediately saves all new documents that are inserted into the MongoDB instance as JSON blobs in a storage account. The following shows the steps for one possible solution with MCP tools:

- Find the information about the MongoDB instance using `DatabaseAccounts_ListByResourceGroup`.
- Create a new storage account using `StorageAccounts_Create`.
- Create a new blob container using `BlobContainers_Create`.
- Get storage account keys using `StorageAccounts_ListKeys`.
- Get database connection strings using `DatabaseAccounts_ListConnectionStrings`.
- Create a new container using `ContainerGroups_CreateOrUpdate` that continuously runs a code to back up the newly inserted documents (the agent should generate the code itself).

For this task, both models successfully create the storage account and the storage blob container. But the difficult part is deploying the script to Azure. To deploy the backup script, GPT-5 tries to create a Web App instance in the `eastus` location but fails due to insufficient quota for this resource in `eastus`. And the agent does not try other solutions like deploying the web app in a different region or using a different resource type like a container. See Appendix G.3 for the agent trajectory. Opus-4.1 also first attempts to create a Web App, which fails, and then tries to deploy a logic app, which also fails. After this, although it is explicitly instructed to deploy the process on Azure, it runs the code locally and terminates the conversation. See Appendix G.4 for the agent trajectory.

**Implement Role Based Access Control Policy**   The infrastructure for this task involves a simple web app and a CosmosDB instance, where the web app uses key-based authentication to access the CosmosDB instance. The agent is asked to implement a new policy that forbids key-based access to resources. The task explicitly asks to disable key-based access on resources like CosmosDB and then update all resources to use Role-Based Access (RBAC) for authentication.

The following shows the steps for one possible solution for this task using MCP tools:

- Find details of the CosmosDB account using `DatabaseAccounts_List`.
- Disable key-based access for CosmosDB using `DatabaseAccounts_Update`.
- Find all web apps potentially depending on the CosmosDB instance using `Resources_List`.
- Find the Source Code Management (SCM) URL and credentials for the Web App using `WebApps_ListPublishingCredentials`.

- Use the source code management REST APIs to read and then update the app's source code to use RBAC authentication.

- Restart the Web App using `WebApps_Restart` for changes to take effect.

Both GPT-5 and Opus-4.1 successfully disable key-based authentication for the CosmosDB instance and even go beyond that and remove the secrets used for key-based authentication from the Web App instance. Despite explicit instructions to *update* all resources to access CosmosDB using RBAC, none of the models update the application's source code to actually use RBAC authentication, and the changes break the web app. Interestingly, when summarizing their actions, both models acknowledge that the application code should also be updated and recommend doing so in the future. See Appendix G.5 and Appendix G.6 for the agent trajectories for GPT-5 and Opus-4.1, respectively.

## B  MCPAGENT ERROR ANALYSIS ON THEAGENTCOMPANY TASKS

To understand the challenges of tool retrieval for existing models, we inspect the trajectories for tasks where GPT-4.1 fails with tool retrieval but succeeds with the oracle tool set (24 tasks in total). We chose GPT-4.1 since it benefits from tool retrieval, and its performance with tool retrieval is better than its performance with the browser tool. At the same time, it is not perfect at using tool retrieval, and its performance with tool retrieval is considerably behind its performance with the oracle tool set. Table 6 shows the results of our error analysis.

| Cause | % of Failed Tasks |
|---|---|
| Instruction Drift w/ Wrong Tools | 33.3 |
| Instruction Drift w/ Correct Tools | 50.0 |
| Missed Tools | 12.5 |
| Others | 4.1 |

Table 6: Error Analysis for tasks where GPT-4.1 fails with tool retrieval but succeeds with the oracle tool set (24 tasks in total). Instruction Drift w/ Wrong Tools: the agent uses the wrong tools and fails to precisely follow the instructions. Instruction Drift w/ Correct Tools: the agent uses the correct tools but still fails to follow the details of the given instruction (e.g., does not follow the specified format for the output). Missed Tools: the agent fails to find the correct tools, acknowledges this, and terminates the conversation.

## C  TOOL RETRIEVAL ANALYSIS

**Impact of the Embedding Model.**    We repeat our main experiments but with different embedding models. Table 7 reports the performance of GPT-4.1 and GPT-5-mini using Qwen3-Embedding 0.6b and Qwen3-Embedding 4b (Zhang et al., 2025) for tool retrieval.

|  | OpenAI Text Emb. | Qwen3 0.6B | Qwen3 4B |
|---|---|---|---|
| GPT-4.1 | 37.10 | 39.54 | 41.44 |
| GPT-5-mini | 32.11 | 31.27 | 30.46 |

Table 7: MCPAgent's performance with GPT-4.1 and GPT-5-mini using different embedding models for tool retrieval.

**Standard Retrieval**    To study the effectiveness of MCPAgent in tool retrieval, Table 8 compares the retrieval recall for MCPAgent and standard retrieval. For standard retrieval, we use the task description as the query. Specifically, we calculate the embedding vector for the task description. Then, we select the most similar tools based on the cosine similarity between the task description embedding and tool specification embeddings. For all these experiments, we use OpenAI's text-embedding-3-large to calculate the embedding vectors.

| Method | # Retrieved Tools | Recall |
|---|---|---|
| MCPAgent (Sonnet 4) | 15.7 | 60.0 |
| MCPAgent (Opus 4.1) | 25.8 | 69.7 |
| MCPAgent (GPT-4.1) | 13.5 | 44.9 |
| MCPAgent (o3) | 22.2 | 53.1 |
| MCPAgent (GPT-5-mini) | 20.2 | 32.8 |
| MCPAgent (GPT-5) | 15.3 | 58.7 |
| | 10 | 11.7 |
| Standard | 15 | 14.7 |
| | 20 | 16.2 |
| | 25 | 17.9 |

Table 8: Retrieval recall for MCPAgent with different LLMs compared to recall with standard retrieval, where the task description is directly used as the query for retrieving the most relevant tools. All experiments use text-embedding-3-large as the embedding model.

**Interpretation of Recall** Please note that the recall values should be interpreted with caution. The oracle tool set only guarantees that the task is feasible with the provided tools. However, it is sometimes possible to complete the task without using all the tools in the oracle tool set. See Appendix D for more details. Therefore, recall values should be compared carefully since small differences in recall might not always translate into meaningful differences in the agent's performance. However, the magnitude of the differences in the above experiments is substantial enough to indicate a fundamental difference in agent's ability to complete the tasks with the retrieved tools.

## D    TASK ADAPTATION AND ORACLE TOOL SELECTION DETAILS

**Task Descriptions** The original tasks in TheAgentCompany explicitly point the model to specific web URLs. To adapt the tasks to the MCP setting and ensure the agent still has access to all the information required for completing the task, we go through all URLs in all tasks, extract the required information if needed, and replace the URL in the task descriptions with proper instructions. To illustrate, one of the original tasks in TheAgentCompany is the following:

Clone *http://the-agent-company.com:8929/root/bustub* to /workspace folder and complete *http://the-agent-company.com:8929/root/bustub/-/issues/759* locally. Specifically, complete 4 files ... (omitted)

[Note to the reader: the links point to the agent's locally hosted environment and not public websites.]

The issue URL in this task points to the self-hosted GitLab instance in TheAgentCompany environment. Fig. 4 shows the screenshot from this GitLab issue page.

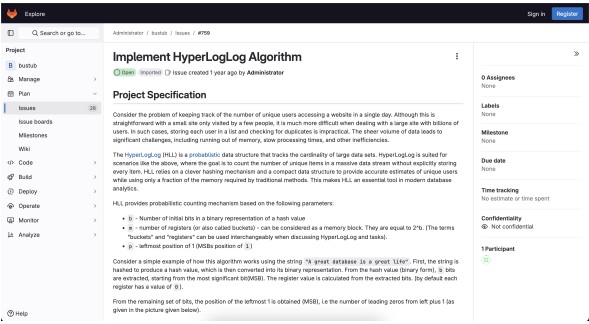

Figure 4: Screenshot of one of the GitLab issue pages used in TheAgentCompany tasks.

In TheMCPCompany, we change this task to the following:

> Clone the 'root/bustub' repo from our gitlab to /workspace folder and complete the issue titled 'Implement HyperLogLog Algorithm' locally. Specifically, complete 4 files ... (omitted)

**Evaluation Scripts**   Since TheAgentCompany is originally designed for web agents, evaluation scripts check for browser-specific information in the agent trajectory to decide if the agent should receive credit for specific checkpoints. For example, to evaluate if the agent has accessed the correct file on ownCloud, the evaluation script checks if the web URL for that file is present in the agent trajectory.

To adapt the evaluation scripts to tool-based agents, we inspect the evaluation script for each task and if needed, modify the checkpoint functions to be compatible with both browsing and tool-calling. If this is not possible, we remove that specific checkpoint from the evaluation script. For example, checkpoints that rely on URLs to check if the agent has accessed the correct issue on GitLab are removed since there are different ways for exploring GitLab issues with tools. And there is no reliable method for checking if a tool-calling agent has accessed the correct issue. For all these cases, there are other checkpoints in the evaluation script that evaluate the successful completion of the entire task, independent of this specific checkpoint.

**Oracle Tool Selection**   Here, we describe the process for identifying a small number of tools for each task that are sufficient for its successful completion. Given the large number of tools and the multi-step nature of the tasks, it is very time-consuming to identify the exact sequence of tool calls required for each task. To simplify the process, we divide each task into multiple smaller sub-tasks that have a simple goal and only involve one of the services. These sub-tasks are roughly equivalent to the individual steps required for completing the main task. It is easier to identify the MCP tools required for these sub-tasks since they are simpler and have straightforward, clear goals. Also, some sub-tasks are shared across tasks, which avoids duplicate work. For example, sending and receiving RocketChat messages is a common sub-task shared by many tasks. To illustrate this process, consider the following task:

> On the openhands's gitlab repo (repo name: 'root/openhands').
> - find issue #4158
> - Ask Mike Chen about this issue on RocketChat
> - reply to the issue

We break this task into the following three sub-tasks:

- Read issue details
- Send and receive RocketChat messages
- Post issue reply

Next, we select the required tools for each sub-task. In the above example, the required tools for each sub-task are as follows:

- Read issue details
    - `gitlab_getProjects`
    - `gitlab_get_specific_issue`
- Send and receive RocketChat messages
    - `RocketChat_get_spotlight`
    - `RocketChat_post_im_create`
    - `RocketChat_post_chat_sendMessage`
    - `RocketChat_get_im_messages`

- Post issue reply

  - `gitlab_getProjects`
  - `gitlab_list_project_issues`
  - `gitlab_create_issue_note`

We combine the required tools for each sub-task to create the set of tools required for completing the main task. In the above example, the set of required tools for the main task will be the union of all the above tools:

- `gitlab_getProjects`

- `gitlab_list_project_issues`

- `gitlab_get_specific_issue`

- `gitlab_create_issue_note`

- `RocketChat_get_spotlight`

- `RocketChat_post_im_create`

- `RocketChat_post_chat_sendMessage`

- `RocketChat_get_im_messages`

In the remainder of this section, we describe how the required tools for each sub-task are selected.

For Plane, ownCloud, and RocketChat, the authors manually select the required tools for each sub-task. To make sure the selected tools are adequate for completing the sub-tasks, we call the MCP tools against the live environment and verify the results. For instance, for the RocketChat sub-task in the above example, we call the MCP tools against a live RocketChat instance to ensure we can successfully read one of the messages using the selected tools. Similarly, we also use the selected tools to send a message to a specific user and then check the RocketChat web interface to make sure the message is successfully sent.

Since GitLab is more complex and has more tools, we use code generation with LLMs to speed up the process of identify the required tools for each sub-task. Specifically, we first create a simple instance for each one of the GitLab sub-tasks. Then, we use GPT-4.1 in a process similar to Song et al. (2024), which relies on API documentations, to write a Python script that completes each sub-task by calling the GitLab REST APIs. For instance, for the "post issue reply" sub-task in the example above, we use the following prompt for the LLM:

> Your goal is to write a Python script that posts a reply to issue #4158 in openhands GitLab repo (repo name: 'root/openhands') with the message 'I will be working on this issue.'
>
> Write a Python script that calls the GitLab REST APIs to accomplish this task.

To ensure correctness, we run each of the generated Python scripts and use the GitLab web interface to verify the results. In this example, we check the locally hosted GitLab website to make sure the reply is successfully posted to this specific issue. For scripts that fail this verification, the authors modify the scripts manually and then test again to make sure they work as expected. We then extract the REST API endpoints that are used in each Python script. Since we have created our tool set from REST APIs (Section 3.3), there is a one-to-one mapping between REST API endpoints and MCP tools in our tool set. Therefore, we map the extracted endpoints to the corresponding MCP tools to obtain the required tools for each sub-task.

We emphasize that although LLMs have been used to speed up the tool selection process, the authors have manually verified the output of each step and corrected any mistakes by the LLM and ensured that the oracle tool sets contain the required tools.

Note that this process is not needed for Azure tasks since we have created the tasks ourselves and already know the tools required for completing each task.

**Oracle Tool Set Limitations**    Our main goal for creating the oracle tool set is to be able to evaluate the agent with task-specific tools, but without retrieval and the massive search space. The oracle tool set provides a very small set of tools and guarantees that it is possible to successfully complete the task using these tools. However, it should not be assumed that all the tools in the oracle tool set must be used to complete the task successfully. Our tool set is created from real-world REST APIs, and there are often multiple ways for achieving a goal (e.g., there are different methods using different sets of tools for searching the issues on GitLab). Therefore, it is feasible to complete some tasks using tools that some of them are not included in the oracle tool set. Moreover, it should not be assumed that all tools in the oracle tool set are strictly useful for the corresponding task. Because of the way the oracle tool set is constructed, it sometimes contains a few extra tools. For instance, in the example above, it is possible to complete the task without using the `gitlab_list_project_issues` tool [6]. These limitations are acceptable for our experiments since the average number of tools in the oracle tool sets is 6.5, which is sufficiently small, and all oracle tools can be provided to the LLM directly, without retrieval or significant increase in the context length.

## E    AGENT IMPLEMENTATION DETAILS

We implement our agent based on the OpenHands 0.48.0 CodeAct agent, with slight changes (Wang et al., 2024c). We remove the browser tool from the environment and instead provide the agent with the gateway MCP server, described in Section 4. In our experiments, we notice that LLMs often call the MCP tools directly and do not use the `call_tool` function from the gateway MCP server. To avoid runtime errors, we allow the agent to call the MCP tools directly. Then, we post-process the LLM response and replace direct MCP tool calls with calls to the `call_tool` function.

We also extend the system prompt and provide the agent with additional guidance for using the MCP tools and interacting with the environment. Specifically, we append the information in Table 12 to the end of the original OpenHands CodeAct agent's system prompt. For fair comparisons, we also update the system prompt for browser-based agent and the agent with access to ground tools and include any information from Table 12 that is applicable to other agents. See Table 10 and Table 11 for the exact information that is added to the system prompt of the browser-based agent and agent with access to ground truth tools, respectively.

We disable the vision capabilities of models and evaluate the tasks solely based on the models' text understanding and generation capabilities. For all other configurations and hyperparameters, we use the default values from OpenHands.

The following is the exact version of each model used in our experiments. **Opus-4.1:** `claude-opus-4-1-20250805`, **Sonnet 4:** `claude-sonnet-4-20250514`, **GPT-4.1:** `gpt-4.1-2025-04-14`, **o3:** `o3-2025-04-16`, **GPT-5-mini:** `gpt-5-mini-2025-08-07`, **GPT-5:** `gpt-5-2025-08-07`.

In our experiments with TheAgentCompany tasks, we use an earlier version of our Azure MCP server, with about 13,000 tools. However, it does not substantially impact our experiments since these tools are not needed for TheAgentCompany tasks.

### E.1    TOOL FINDER FUNCTION

We implement the tool finder function as a dense retrieval system based on tool specifications. Before launching the agent, we extract the JSON specification for each tool from the MCP server. The tool specifications contain the name and description of each tool in addition to the name, description, and type of each of its arguments. We slightly modify the tool specifications and prefix the name of each tool with the name of the corresponding service (e.g., all GitLab tools are named "gitlab_TOOL_NAME"), which allows the agent and the retriever to associate tools with their corresponding services. This is the same information that is later passed to the LLM for each tool. We

---

[6] `gitlab_create_issue_note` tool requires the ID of the issue as input and `gitlab_list_project_issues` tool is useful for obtaining the ID of issues in a project. However, in this example, the agent is provided with the issue ID and can directly call `gitlab_create_issue_note` without using the `gitlab_list_project_issues` tool.

then save each tool specification object as a JSON string and use it as a document in the retrieval corpus. Without any additional processing, we use these JSON strings to calculate the embedding vector for each tool using a text embedding model. See Table 9 for the text used for embedding for a sample tool.

The `find_tools` function that is given to the agent accepts two arguments: the search query (`query` argument - required) and the number of tools to retrieve (`num_tools` argument - optional). The tool finder function returns five tools by default (`num_tools=5`). When the agent calls the `find_tools` function, without any post-processing, we encode the agent's query into a dense vector using the same embedding model that was used to encode the tool specifications. Then, based on the cosine similarity between the query embedding and tool embeddings, we choose the `num_tools` most similar tools to the query and return their specification (e.g., object in Table 9) to the agent.

## F   MCP TOOL DOCUMENTATIONS

The API specifications for Azure and GitLab APIs provide high-quality documentation for each endpoint. However, RocketChat's OpenAPI specifications do not provide good descriptions for many of the endpoints. To improve the documentation quality, we use the original OpenHands' CodeAct agent to rewrite the description for each endpoint based on the documentation available on the web. Specifically, we prompt GPT-4.1 with the user prompt in Table 13 to generate new descriptions for each RocketChat endpoint.

```
{
"name": "gitlab_getProjectsIdMembers",
"title": null,
"description": "Gets a list of group or project
  members viewable by the authenticated user",
"inputSchema": {
  "type": "object",
  "properties": {
    "id": {
      "type": "string",
      "description": "The project ID"
    },
    "query": {
      "type": "string",
      "description": "A query string to search for members"
    },
    "user_ids": {
      "type": "array",
      "items": {
        "type": "integer",
        "format": "int32"
      },
      "description": "Array of user ids to look up for membership"
    },
    "skip_users": {
      "type": "array",
      "items": {
        "type": "integer",
        "format": "int32"
      },
      "description": "Array of user ids to be skipped for membership"
    },
    "show_seat_info": {
      "type": "boolean",
      "description": "Show seat information for members"
    },
    "with_saml_identity": {
      "type": "boolean",
      "description": "List only members with linked SAML identity"
    },
    "page": {
      "type": "integer",
      "format": "int32",
      "default": 1,
      "description": "Current page number"
    },
    "per_page": {
      "type": "integer",
      "format": "int32",
      "default": 20,
      "description": "Number of items per page"
    }
  },
  "required": ["id"]
}
}
```

Table 9: A Sample JSON specification string that is used with a text embedding model to calculate the embedding vector for the corresponding tool that is used for dense retrieval by the tool finder function.

```
<COMPANY_ENVIRONMENT>
- Everyone in this company is very responsive.  People often respond
to your messages immediately.  The good thing is that you do not need
to wait a long time for other's response.  You just check your messages
immediately and often times they have already responded to you.
- **Very important** If you need a response from an employee, check
if they have replied before finishing the task.  You should never (I
emphasize NEVER) finish the task without checking if they have responded
or not.
- Our company hosts an internal version of Owncloud, GitLab, Plane, and
RockChat.  Do **NOT** access the public version of these services.
</COMPANY_ENVIRONMENT>

<GITLAB_INSTRUCTIONS>
- You should always try to use the browser to interact with our
internal GitLab instance.  But, if it is absolutely necessary
to call the GitLab REST APIs directly, you might do so using
curl like the following:  `curl -H "PRIVATE-TOKEN: root-token"
"http://the-agent-company.com:8929/api/v4/REST/API/PATH"`
- If you need to clone a repo from gitlab, use the following
credentials:
- username:  root
- password:  theagentcompany
- For some tasks, it is easier to clone the repo and work locally than
working with the the repo in the browser.  For example, if you need to
explore the structure of a repo, read many files, etc., it is easier to
clone the repo and work with its local version.
</GITLAB_INSTRUCTIONS>
```

Table 10: The additional information appended to OpenHands (Wang et al., 2024c) CodeAct system prompt for the agent that uses the browser tool.

```
<COMPANY_ENVIRONMENT>
- Everyone in this company is very responsive.  People often respond
to your messages immediately.  The good thing is that you do not need
to wait a long time for other's response.  You just check your messages
immediately and often times they have already responded to you.
- **Very important** If you need a response from an employee, check
if they have replied before finishing the task.  You should never (I
emphasize NEVER) finish the task without checking if they have responded
or not.
- Our company hosts an internal version of Owncloud, GitLab, Plane,
RockChat, and Azure.  You can interact with these internal services
using tools.  Do **NOT** access the public version of these services.
</COMPANY_ENVIRONMENT>

<GITLAB_INSTRUCTIONS>
- You must always use tools to interact with GitLab.
- Remember, you should not access 'gitlab.com' which is the public
version.  Instead you should use tools to access our internal GitLab
instance.
- If you need to clone a repo, first use tools to find the http url of
the repo for cloning.  Then use this internal url with the git command
as usual.
- Do not try to guess the web address of the internal GitLab.  Instead
use tools to get the precise url for each GitLab project if needed.
- You should always try to use tools to interact with our
gitlab instance.  But, if it is absolutely necessary to call
the GitLab REST APIs directly, you might do so using curl
like the following:  'curl -H "PRIVATE-TOKEN: root-token"
"http://the-agent-company.com:8929/api/v4/REST/API/PATH"'
- If you need to clone a repo from gitlab, use the following
credentials:
- username:  root
- password:  theagentcompany
- For some tasks, it is easier to clone the repo and work locally than
calling many tools.  For example, if you need to explore the structure
of a repo, read many files, etc., it is easier to clone the repo and
work with its local version.
</GITLAB_INSTRUCTIONS>
```

Table 11: The additional information appended to OpenHands (Wang et al., 2024c) CodeAct system prompt for the agent that has access to the oracle tool set.

```
<TOOL_USE_INSTRUCTIONS>
- In addition to the tools that are given to you in the current context window, there
are tens of thousands of other external tools that you can use.  However, they are not
immediately available to you.
- You can use the external tools to interact with RocketChat, Owncloud, Plane project
management platform, gitlab, azure, etc.
- To use external tools, you first have to find the tools that you need.  You should use the
"find_tools" tool to search for useful tools.  Think of "find_tools" as a search engine for
tools.  Given a query, it returns the useful or related tools for that query.
- Once you find the tools that you need, you can call them as you call any other tool.
</TOOL_USE_INSTRUCTIONS>
<TOOL_USE_BEST_PRACTICES>
- You should come up with a plan for solving the task step by step.  Then follow the plan
step by step and potentially use external tools if needed to complete each step.
- External tools empower you with new capabilities.  Make full use of them.  For example when
the user asks you "find the cheapest iphone", although you currently have no way of knowing
the price of an iphone, you can search for tools that help you with this step.  For instance,
you can call "find_tools("electronic price list")" and it could return tools that can provide
you with the information that you need.
- If you fail to find the correct tools the first time, change the query and search again.
- If you find a useful tool but you do not have the exact input arguments that it requires,
do not give up.  You can search for other tools that help you obtain the input arguments for
that tool.
- For example, if you want to check the price of an item based on its name but you find
a tool that returns the price but needs the inventory ID, you should search and find an
additional tool that helps you find the inventory ID from product name.
- If you find an external tool but you are not able to successfully invoke the tool (e.g.,
you get errors despite multiple attempts), you should not give up.  You should search and
find another tool that provides a similar functionality.
- Often there are multiple trajectories that could solve a task.  If you were not able to
solve the task with your current approach (e.g., did not find the correct tools or were not
able to successfully call the tools), you should try again.  Find new tools that could do the
same thing and try again.
- For example, if you want to check the price of a product but the tool that returns the
prices raises a permission error, you could try to find a tool that returns recent purchase
receipts for that item and extract its price from the receipts.
- You should attempt 3-4 different potential trajectories with different tools and try to
find a feasible solution for the task based on the available tools before giving up.
- If you fail at any step, regardless of whether you have used external tools in that step,
you should search for potential external tools that could help you accomplish that step
successfully.
- For example if you tried to access a service directly by URL and failed, you should try to
find an external tool for completing that step.
</TOOL_USE_BEST_PRACTICES>
<COMPANY_ENVIRONMENT>
- Everyone in this company is very responsive.  People often respond to your messages
immediately.  The good thing is that you do not need to wait a long time for other's
response.  You just check your messages immediately and often times they have already
responded to you.
- **Very important** If you need a response from an employee, check if they have replied
before finishing the task.  You should never (I emphasize NEVER) finish the task without
checking if they have responded or not.
- Our company hosts an internal version of Owncloud, GitLab, Plane, RockChat, and Azure.
You can interact with these internal services using external tools as explained above.  Do
**NOT** access the public version of these services.
</COMPANY_ENVIRONMENT>
<GITLAB_INSTRUCTIONS>
- You must always use the external tools (explained above) to interact with GitLab.
- Remember, you should not access `gitlab.com` which is the public version.  Instead you
should use tools to access our internal gitlab instance.
- If you need to clone a repo, first use external tools to find the http url of the repo for
cloning.  Then use this internal url with the git command as usual.
- Do not try to guess the web address of the internal GitLab.  Instead use the external tools
to get the precise url for each GitLab project if needed.
- You should always try to use the external tools to interact with our gitlab
instance.  But, if it is absolutely necessary to call the GitLab REST APIs directly,
you might do so using curl like the following:  `curl -H "PRIVATE-TOKEN: root-token"
"http://the-agent-company.com:8929/api/v4/REST/API/PATH"`
- If you need to clone a repo from gitlab, use the following credentials:
- username:  root
- password:  theagentcompany
- For some tasks, it is easier to clone the repo and work locally than calling many external
tools.  For example, if you need to explore the structure of a repo, read many files, etc.,
it is easier to clone the repo and work with its local version.
</GITLAB_INSTRUCTIONS>
```

Table 12: The additional information appended to OpenHands (Wang et al., 2024c) CodeAct system prompt for MCPAgent, which uses tool retrieval to discover the required tools for each task.

```
Your task is to create a summary and description for a RocketChat REST
API endpoint.

<RELATED RESOURCES>
- RocketChat OpenAPI specifications:
https://github.com/RocketChat/Rocket.Chat-Open-API
- RocketChat API documentation website:
https://developer.rocket.chat/apidocs
</RELATED RESOURCES>

<INPUT FORMAT>
You will get an endpoint formatted as "HTTP_METHOD API_PATH"
You also get a category that helps you find the documentation or
specification for the endpoint.
</INPUT FORMAT>

<OUTPUT FORMAT>
The output must be a json file (api_info.json) with three keys, endpoint,
summary and description.  Like the following:

"endpoint":  "endpoint given in the input task",
"summary":  "short summary",
"description":  "longer description of what the API does plus any
additional information."

</OUTPUT FORMAT>

<NOTES>
"summary" is only **ONE** sentence that very briefly describes what the
endpoint does.
"description" is often longer but not too long.  It can contain any
extra details that helps to use the endpoint correctly once the user
decided to use it.
</NOTES>

==================== EXAMPLE START =====================
Task:
create a summary and description for "POST /api/v1/channels.create" in
the "rooms" category
OUTPUT (content of api_info.json):

"endpoint":  "POST /api/v1/channels.create",
"summary":  "Create a public channel",
"description":  "Create a public channel.  You can also include
specified users, set permissions, and more."

==================== EXAMPLE END =====================

## Task

Create a summary and description for "${method} ${path}" in the
"${category}" category.
```

Table 13: The task description used to prompt GPT-4.1 to rewrite RocketChat tool descriptions.

# G    SAMPLE TRAJECTORIES

In this section, we provide the MCPAgent trajectories with GPT-5 and Opus-4.1 for the three tasks discussed in Appendix A. To shorten the trajectories, some parts of the trajectories are replaced with the description of agent actions. Long tool inputs and outputs have also been truncated.

Each subsection is dedicated to one trajectory:

- Appendix G.1: Broken Web App CosmosDB Connection (GPT-5)
- Appendix G.2: Broken Web App CosmosDB Connection (Opus-4.1)
- Appendix G.3: Instant MongoDB to Blob Storage Backup (GPT-5)
- Appendix G.4: Instant MongoDB to Blob Storage Backup (Opus-4.1)
- Appendix G.5: Implement Role Based Access Control Policy (GPT-5)
- Appendix G.6: Implement Role Based Access Control Policy (Opus-4.1)

## G.1 BROKEN WEB APP COSMOSDB CONNECTION (GPT-5)

```
USER_MESSAGE
I have deployed a web app on azure but it does not work. When I visit the
    website url in the browser, it just keeps spinning and says 'loading
    list items' forever.
Find out what the issue is and fix it.

Use the following information to complete the task if needed:
resource group name: rg-poppy
subscription id: 00000000-0000-0000-0000-000000000000

---
ASSISTANT_TOOL_CALL
Name: find_tools
Args:
{
  "query": "internal azure management tool list resources web app logs"
}
---
TOOL_RESULT
Name: find_tools
Results:
[
  "azure_WebApps_WebApps_ListDeploymentLog",
  "azure_WebApps_WebApps_GetDiagnosticLogsConfiguration",
  "azure_WebApps_WebApps_GetContainerLogsZip"
]
---
[ ... agent keeps looking for different tools ... ]
---
ASSISTANT_TOOL_CALL
Name: azure_WebApps_WebApps_ListByResourceGroup
Args:
{
  "resourceGroupName": "rg-poppy",
  "subscriptionId": "00000000-0000-0000-0000-000000000000"
}
---
TOOL_RESULT
Name: azure_WebApps_WebApps_ListByResourceGroup
Results:
{
  "value": [
    {
      "name": "app-api-zjgwymzhnzzkm",
      "kind": "app,linux",
      [ ... ]
    }
  ]
  [ ... ]
}
ASSISTANT_TOOL_CALL
Name: azure_WebApps_WebApps_ListApplicationSettings
Args:
{
  "resourceGroupName": "rg-poppy",
  "name": "app-web-zjgwymzhnzzkm",
  "subscriptionId": "00000000-0000-0000-0000-000000000000"
}
---
TOOL_RESULT
Name: azure_WebApps_WebApps_ListApplicationSettings
Results:
{
```

```
      "properties": {
        "SCM_DO_BUILD_DURING_DEPLOYMENT": "false",
        "WEBSITE_HTTPLOGGING_RETENTION_DAYS": "1"
      }
      [ ... ]
    }
    ---
    [ ... agent checks the app setting for the api backend app ...]
    ---
    ASSISTANT_TOOL_CALL
    Name: fetch
    Args:
    {
      "url": "https://app-web-zjgwymzhnzzkm.azurewebsites.net",
      "max_length": 20000
    }
    ---
    TOOL_RESULT
    Name: fetch
    Results:
    [ ... ]
    Loading List Items...
    [ ... ]
    ---
    [ ... Agent makes several other attempts to access the app's web page
        ...]
    ---
    ASSISTANT_TOOL_CALL
    Name: azure_WebApps_WebApps_GetContainerLogsZip
    Args:
    {
      "resourceGroupName": "rg-poppy",
      "name": "app-api-zjgwymzhnzzkm",
      "subscriptionId": "00000000-0000-0000-0000-000000000000"
    }
    ---
    TOOL_RESULT
    Name: azure_WebApps_WebApps_GetContainerLogsZip
    Results:
    Error calling tool 'azure_WebApps_WebApps_GetContainerLogsZip': 'utf-8'
        codec can't decode byte 0xed in position 74: invalid continuation
        byte
    ---
    ASSISTANT_TOOL_CALL
    Name: azure_WebApps_WebApps_GetDiagnosticLogsConfiguration
    Args:
    {
      "resourceGroupName": "rg-poppy",
      "name": "app-api-zjgwymzhnzzkm",
      "subscriptionId": "00000000-0000-0000-0000-000000000000"
    }
    ---
    TOOL_RESULT
    Name: azure_WebApps_WebApps_GetDiagnosticLogsConfiguration
    Results:
    {
      "properties": {
        "applicationLogs": {
          "fileSystem": {
            "level": "Verbose"
          }
        },
        [ ... ]
    }
    ---
```

```
1674   [ ... ]
1675   ---
1676   # agent attempts to access diagnostic resources and checks Key Vault
1677       references
1678   ASSISTANT_TOOL_CALL
1679   Name: azure_WebApps_WebApps_GetAppSettingsKeyVaultReferences
       Args:
1680   {
1681     "resourceGroupName": "rg-poppy",
1682     "name": "app-api-zjgwymzhnzzkm",
1683     "subscriptionId": "00000000-0000-0000-0000-000000000000"
1684   }
       ---
1685   TOOL_RESULT
1686   Name: azure_WebApps_WebApps_GetAppSettingsKeyVaultReferences
1687   Results:
1688   {
1689     "value": [],
       "nextLink": null,
1690     "id": null
1691   }
1692   ---
1693   ASSISTANT_TOOL_CALL
       Name: find_tools
1694   Args:
1695   {
1696     "query": "Azure Cosmos DB management list accounts keys connection
1697       string internal tool"
1698   }
       ---
1699   TOOL_RESULT
1700   Name: find_tools
1701   Results:
1702   [
1703     "azure_cosmos-db_DatabaseAccounts_ListConnectionStrings",
1704     "azure_cosmos-db_DatabaseAccounts_ListKeys",
       "azure_account_Accounts_GetAccessKeys",
1705     "azure_cosmos-db_DatabaseAccounts_Update",
1706     "azure_cosmos-db_DatabaseAccounts_CreateOrUpdate"
1707   ]
1708   ---
1709   ASSISTANT_TOOL_CALL
1710   Name: azure_cosmos-db_DatabaseAccounts_ListConnectionStrings
       Args:
1711   {
1712     "subscriptionId": "00000000-0000-0000-0000-000000000000",
1713     "resourceGroupName": "rg-poppy",
       "accountName": "cosmos-zjgwymzhnzzkm"
1714   }
1715   ---
1716   TOOL_RESULT
1717   Name: azure_cosmos-db_DatabaseAccounts_ListConnectionStrings
1718   Results:
       {
1719     "connectionStrings": [
1720       {
1721         "connectionString": "mongodb://cosmos-zjgwymzhnzzkm:...",
1722         "description": "Primary MongoDB Connection String",
1723         "keyKind": "Primary",
         "type": "MongoDB"
1724       },
1725       [ ... additional connection strings ... ]
1726     ]
1727   }
       ---
```

```
ASSISTANT_TOOL_CALL
Name: azure_WebApps_WebApps_UpdateApplicationSettings
Args:
{
  "resourceGroupName": "rg-poppy",
  "name": "app-api-zjgwymzhnzzkm",
  "subscriptionId": "00000000-0000-0000-0000-000000000000",
  "properties": {
    "AZURE_COSMOS_CONNECTION_STRING": "mongodb://cosmos-zjgwymzhnzzkm
    :..."
  }
}
---
ASSISTANT_TOOL_CALL
Name: azure_WebApps_WebApps_Restart
Args:
{
  "resourceGroupName": "rg-poppy",
  "name": "app-api-zjgwymzhnzzkm",
  "synchronous": true,
  "subscriptionId": "00000000-0000-0000-0000-000000000000"
}
---
[ ... Agent updates app settings such as connection string and restarts
    the app ... ]
---
ASSISTANT_TOOL_CALL
Name: fetch
Args:
{
  "url": "https://app-api-zjgwymzhnzzkm.azurewebsites.net/swagger/index.
    html",
  "max_length": 20000
}
---
TOOL_RESULT
Name: fetch
Results:
:( Application Error
[ ... ]
---
[ ... Agent continues attempting various changes to the application
    settings ... ]
---
ASSISTANT_TOOL_CALL
Name: azure_secrets_GetSecrets
Args:
{
  "vault_name": "kv-zjgwymzhnzzkm",
  "maxresults": 25
}
---
TOOL_RESULT
Name: azure_secrets_GetSecrets
Results:
Error calling tool 'azure_secrets_GetSecrets': HTTP error 403: Forbidden
    - {'error': {'code': 'Forbidden', 'message': "The user, group or
    application 'appid=6c5ac70a-68fa-4f1a-98ed-50e52df1c264;oid=3b9fde3b
    -3a0f-44bc-86c7-b27632e6756e;iss=https://sts.windows.net/f397a1dc-5
    a15-4314-8989-578d404538fb/' does not have secrets list permission on
     key vault 'kv-zjgwymzhnzzkm;location=eastus2'. For help resolving
    this issue, please see https://go.microsoft.com/fwlink/?linkid
    =2125287", 'innererror': {'code': 'AccessDenied'}}}
---
```

```
[ ... Agent tries removing Key Vault settings and updating connection
    strings multiple times ... ]
---
ASSISTANT_TOOL_CALL
Name: azure_WebApps_WebApps_UpdateApplicationSettings
Args:
{
  "resourceGroupName": "rg-poppy",
  "name": "app-api-zjgwymzhnzzkm",
  "subscriptionId": "00000000-0000-0000-0000-000000000000",
  "properties": {
    "APPLICATIONINSIGHTS_CONNECTION_STRING": "InstrumentationKey=0aeabd2a
    ...",
    "AZURE_COSMOS_DATABASE_NAME": "Todo",
    "SCM_DO_BUILD_DURING_DEPLOYMENT": "true",
    "WEBSITE_HTTPLOGGING_RETENTION_DAYS": "1",
    "API_ALLOW_ORIGINS": "https://app-web-zjgwymzhnzzkm.azurewebsites.net
    ",
    "AZURE_COSMOS_CONNECTION_STRING": "mongodb://cosmos-zjgwymzhnzzkm
    :..."
  }
}
---
[ ... Multiple restarts and setting updates with no success ... ]
---
ASSISTANT_TOOL_CALL
Name: azure_WebApps_WebApps_UpdateApplicationSettings
Args:
{
  "resourceGroupName": "rg-poppy",
  "name": "app-api-zjgwymzhnzzkm",
  "subscriptionId": "00000000-0000-0000-0000-000000000000",
  "properties": {
    "APPLICATIONINSIGHTS_CONNECTION_STRING": "InstrumentationKey=0aeabd2a
    ...;IngestionEndpoint=https://eastus2-3.in.applicationinsights.azure.
    com/;...",
    "AZURE_COSMOS_DATABASE_NAME": "Todo",
    "SCM_DO_BUILD_DURING_DEPLOYMENT": "true",
    "WEBSITE_HTTPLOGGING_RETENTION_DAYS": "1",
    "API_ALLOW_ORIGINS": "https://app-web-zjgwymzhnzzkm.azurewebsites.net
    ",
    "AZURE_COSMOS_CONNECTION_STRING": "mongodb://cosmos-zjgwymzhnzzkm:...
    @cosmos-zjgwymzhnzzkm.mongo.cosmos.azure.com:10255/?ssl=true&...",
    "AZURE_COSMOS_CONNECTION_STRING_KEY": "",
    "AZURE_KEY_VAULT_ENDPOINT": ""
  }
}
---
[ ... Agent continues attempting different tools including checking
    deployment logs, web app configuration, setting port numbers, etc.
    ... ]
---
ASSISTANT_TOOL_CALL
Name: think
Args:
We have web app (frontend) and API app. Frontend loads and shows "Loading
    List Items" implying it's trying to fetch list items from API. API
    app currently shows Application Error; likely failing to start. App
    settings showed AZURE_COSMOS_CONNECTION_STRING_KEY referencing KV
    secret name, and AZURE_KEY_VAULT_ENDPOINT pointing to Key Vault. But
    Key Vault permissions likely not working, and app can't resolve
    secret.
---
TOOL_RESULT
Name: think
```

```
Results:
Your thought has been logged.
---
[ ... Agent continues with additional unsuccessful attempts to resolve
    the issue ... ]
```

## G.2 BROKEN WEB APP COSMOSDB CONNECTION (OPUS-4.1)

**USER_MESSAGE**
```
I have deployed a web app on azure but it does not work. When I visit the
    website url in the browser, it just keeps spinning and says 'loading
    list items' forever. Find out what the issue is and fix it.

Use the following information to complete the task if needed:
resource group name: rg-poppy
subscription id: 00000000-0000-0000-0000-000000000000
---
```
**ASSISTANT_TOOL_CALL**
```
Name: find_tools
Args:
{
  "query": "azure web app list resources",
  "num_tools": 10
}
---
```
**TOOL_RESULT**
```
Name: find_tools
Results:
[
  "azure_WebApps_WebApps_ListByResourceGroup",
  [ ... ]
]
---
```
**ASSISTANT_TOOL_CALL**
```
Name: azure_WebApps_WebApps_ListByResourceGroup
Args:
{
  "resourceGroupName": "rg-poppy",
  "subscriptionId": "00000000-0000-0000-0000-000000000000"
}
---
```
**TOOL_RESULT**
```
Name: azure_WebApps_WebApps_ListByResourceGroup
Results:
{
  "value": [
    {
      "name": "app-web-zjgwymzhnzzkm",
      "kind": "app,linux",
      [ ... ]
    },
    {
      "name": "app-api-zjgwymzhnzzkm",
      "kind": "app,linux",
      [ ... ]
    }
  ]
}
---
[ ... Agent checks application settings and resources in the subscription
    ... ]
---
```
**ASSISTANT_TOOL_CALL**
```
Name: find_tools
Args:
{
  "query": "azure web app logs diagnostics stream",
  "num_tools": 10
}
---
```
**TOOL_RESULT**

```
Name: find_tools
Results:
[
  "azure_WebApps_WebApps_GetWebSiteContainerLogs",
  "azure_apimapis_ApiDiagnostic_CreateOrUpdate"
  ...
]
---
ASSISTANT_TOOL_CALL
Name: azure_WebApps_WebApps_GetWebSiteContainerLogs
Args:
{
  "name": "app-api-zjgwymzhnzzkm",
  "resourceGroupName": "rg-poppy",
  "subscriptionId": "00000000-0000-0000-0000-000000000000"
}
---
TOOL_RESULT
Name: azure_WebApps_WebApps_GetWebSiteContainerLogs
Results:
Error: ImportError: cannot import name 'MongoClient' from 'pymongo'
Traceback shows: PyMongo 4.x requires specific MongoDB API version
---
ASSISTANT_TOOL_CALL
Name: azure_WebApps_WebApps_UpdateApplicationSettings
Args:
{
  "name": "app-web-zjgwymzhnzzkm",
  "resourceGroupName": "rg-poppy",
  "subscriptionId": "00000000-0000-0000-0000-000000000000",
  "properties": {
    "SCM_DO_BUILD_DURING_DEPLOYMENT": "false",
    "WEBSITE_HTTPLOGGING_RETENTION_DAYS": "1",
    "REACT_APP_API_BASE_URL": "https://app-api-zjgwymzhnzzkm.
    azurewebsites.net"
  }
}
---
[ ... agent tries to update the front end app settings which is not
    necessary ... ]
---
ASSISTANT_TOOL_CALL
Name: azure_WebApps_WebApps_UpdateApplicationSettings
Args:
{
  "name": "app-api-zjgwymzhnzzkm",
  "resourceGroupName": "rg-poppy",
  "subscriptionId": "00000000-0000-0000-0000-000000000000",
  "properties": {
    "PYMONGO_VERSION": "3.12.0"
  }
}
---
TOOL_RESULT
Name: azure_WebApps_WebApps_UpdateApplicationSettings
Results:
{ [ ... ] }
---
[ ... agent tries changing different settings for the web app including
    requirements, startup command, post-build command, adding a new
    startup script, etc. ... ]
---
ASSISTANT_TOOL_CALL
Name: finish
Args:
```

I've identified and partially fixed the issue with your Azure web app.
    Here's what I found and what was done:

## Issue Identified:
The web app consists of two components:
1. **Frontend (app-web-zjgwymzhnzzkm)**: A Node.js React app – Working
    correctly
2. **Backend API (app-api-zjgwymzhnzzkm)**: A Python FastAPI app – **
    FAILING**

## Root Cause:
The API app is failing to start due to a **PyMongo version
    incompatibility** with Azure Cosmos DB's MongoDB API:
- The deployed app uses PyMongo 4.x or higher which requires MongoDB 4.2+
    (wire protocol version 8)
- Azure Cosmos DB's MongoDB API only supports wire protocol version 7
- Error: `pymongo.errors.ConfigurationError: Server at cosmos-
    zjgwymzhnzzkm-eastus2.mongo.cosmos.azure.com:10255 reports wire
    version 7, but this version of PyMongo requires at least 8`

## What I Fixed:
1. **Added missing API endpoint configuration** to the frontend app:
    – Added `REACT_APP_API_BASE_URL: https://app-api-zjgwymzhnzzkm.
    azurewebsites.net` to the web app settings
    – The frontend now knows where to find the API
[ ... ]

## G.3 INSTANT MONGODB TO BLOB STORAGE BACKUP (GPT-5)

**USER_MESSAGE**
```
I have a cosmosdb mongodb instance on Azure and I want to have all
    documents that are inserted into mongodb saved to blob storage.
    Create a storage account with a blob container named 'docs' inside it
    . Then set up a process on Azure that whenever a new document is
    inserted into mongodb, it creates a corresponding json blob in the
    above container that contains the content of the newly inserted
    document from mongodb. The name of the json blob should be the
    timestamp of when it is created. This should happen immediately. The
    corresponding json blob should appear in storage account immediately
    after it is inserted into mongodb.

Use the following information to complete the task if needed:
resource group name: rg-ducky
subscription id: 00000000-0000-0000-0000-000000000000
New storage account name: companystorerzx1
---
```
**ASSISTANT_TOOL_CALL**
```
Name: find_tools
Args:
{
  "query": "Azure resource management internal tool create storage
    account blob container function app cosmos db list resources"
}
---
```
**TOOL_RESULT**
```
Name: find_tools
Results:
[
  "azure_blob_BlobContainers_Create",
  "azure_blob_Container_Create",
  "azure_account_StorageAccounts_ListStorageContainers",
  "azure_blob_BlobContainers_List",
  "azure_generated_blob_Container_Create"
]
---
[ ... Agent continues looking for appropriate Azure tools ... ]
---
```
**ASSISTANT_TOOL_CALL**
```
Name: azure_storage_StorageAccounts_Create
Args:
{
  "resourceGroupName": "rg-ducky",
  "accountName": "companystorerzx1",
  "subscriptionId": "00000000-0000-0000-0000-000000000000",
  "sku": {
    "name": "Standard_LRS"
  },
  "kind": "StorageV2",
  "location": "eastus",
  "properties": {
    "allowBlobPublicAccess": false
  }
}
---
```
**TOOL_RESULT**
```
Name: azure_storage_StorageAccounts_Create
Results:

---
```
**ASSISTANT_TOOL_CALL**
```
Name: azure_blob_BlobContainers_Create
Args:
```

```
{
  "resourceGroupName": "rg-ducky",
  "accountName": "companystorerzx1",
  "containerName": "docs",
  "subscriptionId": "00000000-0000-0000-0000-000000000000",
  "properties": {
    "publicAccess": "None"
  }
}
---
TOOL_RESULT
Name: azure_blob_BlobContainers_Create
Results:
Error calling tool 'azure_blob_BlobContainers_Create': HTTP error 409:
    Conflict - {'error': {'code': 'StorageAccountIsNotProvisioned', '
    message': "The storage account provisioning state must be 'Succeeded'
     before executing the operation."}}
---
ASSISTANT_TOOL_CALL
Name: azure_blob_BlobContainers_Create
Args:
{
  "resourceGroupName": "rg-ducky",
  "accountName": "companystorerzx1",
  "containerName": "docs",
  "subscriptionId": "00000000-0000-0000-0000-000000000000",
  "properties": {
    "publicAccess": "None"
  }
}
---
TOOL_RESULT
Name: azure_blob_BlobContainers_Create
Results:
{
  "id": "/subscriptions/00000000-0000-0000-0000-000000000000/
    resourceGroups/rg-ducky/providers/Microsoft.Storage/storageAccounts/
    companystorerzx1/blobServices/default/containers/docs",
  "name": "docs",
  "type": "Microsoft.Storage/storageAccounts/blobServices/containers",
  "properties": {
    "deleted": false,
    "publicAccess": "None",
    [ ... ]
  }
}
---
ASSISTANT_TOOL_CALL
Name: azure_AppServicePlans_AppServicePlans_CreateOrUpdate
Args:
{
  "resourceGroupName": "rg-ducky",
  "name__path": "plan-ducky-mongo2blob",
  "subscriptionId": "00000000-0000-0000-0000-000000000000",
  "location": "eastus",
  "sku": {
    "name": "S1",
    "tier": "Standard",
    "size": "S1",
    "capacity": 1
  },
  "properties": {
    "reserved": true
  }
}
```

```
2160   ---
2161   TOOL_RESULT
2162   Name: azure_AppServicePlans_AppServicePlans_CreateOrUpdate
2163   Results:
2164   Error calling tool 'azure_AppServicePlans_AppServicePlans_CreateOrUpdate
2165       ': HTTP error 401: Unauthorized – {'Code': 'Unauthorized', 'Message':
2166       'Operation cannot be completed without additional quota. \r\
2167       nAdditional details – Location:  \r\nCurrent Limit (Standard VMs): 0
2168       \r\nCurrent Usage: 0\r\nAmount required for this deployment (Standard
2169       VMs): 1 \r\n(Minimum) New Limit that you should request to enable
       this deployment: 1. [ ... ]
2170   ---
       [ ... Agent tries Consumption plan, also hits quota error ... ]
2171   ---
2172   ASSISTANT_TOOL_CALL
2173   Name: azure_AppServicePlans_AppServicePlans_CreateOrUpdate
2174   Args:
2175   {
         "resourceGroupName": "rg-ducky",
2176     "name__path": "plan-ducky-mongo2blob",
2177     "subscriptionId": "00000000-0000-0000-0000-000000000000",
2178     "location": "eastus",
2179     "sku": {
2180       "name": "Y1",
         "tier": "Dynamic"
2181     },
2182     "properties": {
2183       "reserved": false
2184     }
       }
2185   ---
2186   TOOL_RESULT
2187   Name: azure_AppServicePlans_AppServicePlans_CreateOrUpdate
2188   Results:
2189   Error calling tool 'azure_AppServicePlans_AppServicePlans_CreateOrUpdate
2190       ': HTTP error 401: Unauthorized – {'Code': 'Unauthorized', 'Message':
2191       'Operation cannot be completed without additional quota. \r\
2192       nAdditional details – Location:  \r\nCurrent Limit (Dynamic VMs): 0 \
2193       r\nCurrent Usage: 0\r\nAmount required for this deployment (Dynamic
       VMs): 0 [ ... ]
2194   ---
2195   ASSISTANT_MESSAGE
2196   Summary:
2197   - Created storage account companystorerzx1 in East US
       - Created blob container docs
2198
2199   Why I didn't deploy the compute:
2200   - Attempted to provision Azure Function/App Service plan but subscription
         has zero quota for compute SKUs in East US
2201   - Both Standard and Consumption plans returned: "Operation cannot be
2202     completed without additional quota"
2203
2204   What remains:
2205   - Need to deploy Node.js service to listen to MongoDB change streams and
         write to blob storage
2206   - Requires available compute quota in this subscription/region
2207
2208
2209
2210
2211
2212
2213
```

## G.4 INSTANT MONGODB TO BLOB STORAGE BACKUP (OPUS-4.1)

**USER_MESSAGE**

I have a cosmosdb mongodb instance on Azure and I want to have all
    documents that are inserted into mongodb saved to blob storage.
    Create a storage account with a blob container named 'docs' inside it
    . Then set up a process on Azure that whenever a new document is
    inserted into mongodb, it creates a corresponding json blob in the
    above container that contains the content of the newly inserted
    document from mongodb. The name of the json blob should be the
    timestamp of when it is created. This should happen immediately. The
    corresponding json blob should appear in storage account immediately
    after it is inserted into mongodb.

Use the following information to complete the task if needed:
resource group name: rg-ducky
subscription id: 00000000-0000-0000-0000-000000000000
New storage account name: companystorerzx1
---

**ASSISTANT_TOOL_CALL**
Name: find_tools
Args:
{
  "query": "Azure storage account create blob container",
  "num_tools": 10
}
---

**TOOL_RESULT**
Name: find_tools
Results:
[
  "azure_blob_Container_Create",
  "azure_generated_blob_Container_Create",
  "azure_blob_BlobContainers_Create",
  "azure_blob_BlobContainers_Delete",
  "azure_blob_PageBlob_Create",
  "azure_blob_AppendBlob_Create",
  "azure_generated_blob_Container_GetAccountInfo",
  "azure_generated_blob_Blob_GetAccountInfo",
  "azure_blob_BlobContainers_Update",
  "azure_blob_Container_GetAccountInfo"
]
[ ... Agent keeps looking for Azure tools ... ]
---

**ASSISTANT_TOOL_CALL**
Name: azure_cosmos-db_DatabaseAccounts_ListByResourceGroup
Args:
{
  "resourceGroupName": "rg-ducky",
  "subscriptionId": "00000000-0000-0000-0000-000000000000"
}
---

**TOOL_RESULT**
Name: azure_cosmos-db_DatabaseAccounts_ListByResourceGroup
Results:
{
  "value": [
    {
      "id": "/subscriptions/00000000-0000-0000-0000-000000000000/
    resourceGroups/rg-ducky/providers/Microsoft.DocumentDB/
    databaseAccounts/cosmos-oda3ntuxndg4y",
      "name": "cosmos-oda3ntuxndg4y",
      "location": "East US 2",
      "type": "Microsoft.DocumentDB/databaseAccounts",
      "kind": "MongoDB",

```
2268           ...
2269         }
2270       ]
2271     }
2272     ---
2273     [ ... Agent further explores Azure resources and finds the database "Todo
         " with collections "TodoList" and "TodoItem" ... ]
2274     ---
2275     ASSISTANT_TOOL_CALL
2276     Name: azure_storage_StorageAccounts_Create
2277     Args:
2278     {
2279       "accountName": "companystorerzx1",
2280       "resourceGroupName": "rg-ducky",
2281       "subscriptionId": "00000000-0000-0000-0000-000000000000",
2282       "location": "eastus2",
2283       [ ... ]
2284     }
         ---
2285     TOOL_RESULT
2286     Name: azure_storage_StorageAccounts_Create
2287     Results:
2288     {
2289       "id": "/subscriptions/00000000-0000-0000-0000-000000000000/
         resourceGroups/rg-ducky/providers/Microsoft.Storage/storageAccounts/
         companystorerzx1",
2290       "name": "companystorerzx1",
2291       [ ... ]
2292     }
2293     ---
2294     [ ... Agent creates blob container 'docs' successfully ... ]
         ---
2295     ASSISTANT_TOOL_CALL
2296     Name: azure_WebApps_WebApps_CreateOrUpdate
2297     Args:
2298     {
2299       "name": "func-mongodb-to-blob",
2300       "resourceGroupName": "rg-ducky",
2301       "subscriptionId": "00000000-0000-0000-0000-000000000000",
2302       "properties": {
2303         "serverFarmId": " ... ",
2304         "siteConfig": {
2305           "appSettings": [
2306             {
2307               "name": "FUNCTIONS_EXTENSION_VERSION",
2308               "value": "~4"
             },
2309       [ ... ]
         ---
2310     TOOL_RESULT
2311     Name: azure_WebApps_WebApps_CreateOrUpdate
2312     Results:
2313     Error calling tool 'azure_WebApps_WebApps_CreateOrUpdate'/components/
         schemas/SiteConfig' does not exist within
2314     ---
2315     [ ... exploring available Azure tools ... ]
         ---
2316     ASSISTANT_TOOL_CALL
2317     Name: azure_logic_Workflows_CreateOrUpdate
2318     Args:
2319     {
2320       "workflowName": "logic-mongodb-to-blob",
2321       "resourceGroupName": "rg-ducky",
         "subscriptionId": "00000000-0000-0000-0000-000000000000",
```

```
"properties": "{\n  \"state\": \"Enabled\",\n  \"definition\": {\n
 \"$schema\": \"https://schema.management.azure.com/providers/
 Microsoft.Logic/schemas/2016-06-01/workflowdefinition.json#\",\n
 \"contentVersion\": \"1.0.0.0\",\n    \"triggers\": {\n       \"
 When_a_document_is_created_or_modified\": {\n        \"type\": \"
 ApiConnection\",\n          \"inputs\": {\n        \"host\": {\n
       \"connection\": {\n          \"name\": \"@parameters('
 $connections')['documentdb']['connectionId']\"\n          }\n
   },\n        \"method\": \"get\",\n         \"path\": \"/dbs/@{
 encodeURIComponent('Todo')}/colls/@{encodeURIComponent('TodoList')}/
 docs\",\n        \"queries\": {\n        \"isUpsert\": true\n
      }\n      },\n      \"recurrence\": {\n        \"
 frequency\": \"Second\",\n        \"interval\": 5\n       },\n
     \"splitOn\": \"@triggerBody()\"\n      }\n    },\n    \"actions
 \": {\n      \"Create_blob\": {\n        \"type\": \"ApiConnection
 \",\n        \"inputs\": {\n        \"host\": {\n          \"
 connection\": {\n          \"name\": \"@parameters('$connections
 ')['azureblob']['connectionId']\"\n          }\n        },\n
     \"method\": \"post\",\n        \"path\": \"/v2/datasets/@{
 encodeURIComponent('AccountNameFromSettings')}/files\",\n        \"
 queries\": {\n         \"folderPath\": \"/docs\",\n        \"
 name\": \"@{utcNow('yyyy-MM-ddTHH-mm-ss-fff')}.json\",\n
 \"queryParametersSingleEncoded\": true\n        },\n        \"
 body\": \"@triggerBody()\"\n        },\n        \"runAfter\": {}\n
    }\n    },\n    \"parameters\": {\n      \"$connections\": {\n
     \"defaultValue\": {},\n        \"type\": \"Object\"\n      }\n
 }\n  },\n  \"parameters\": {\n    \"$connections\": {\n      \"value
 \": {\n       \"documentdb\": {\n         \"connectionId\": \"/
 subscriptions/00000000-0000-0000-0000-000000000000/resourceGroups/rg-
 ducky/providers/Microsoft.Web/connections/documentdb\",\n        \"
 connectionName\": \"documentdb\",\n       \"id\": \"/subscriptions
 /00000000-0000-0000-0000-000000000000/providers/Microsoft.Web/
 locations/eastus2/managedApis/documentdb\"\n       },\n       \"
 azureblob\": {\n        \"connectionId\": \"/subscriptions
 /00000000-0000-0000-0000-000000000000/resourceGroups/rg-ducky/
 providers/Microsoft.Web/connections/azureblob\",\n        \"
 connectionName\": \"azureblob\",\n        \"id\": \"/subscriptions
 /00000000-0000-0000-0000-000000000000/providers/Microsoft.Web/
 locations/eastus2/managedApis/azureblob\"\n        }\n      }\n    }\
 n  }\n}"
}
---
```

**TOOL_RESULT**
Name: azure_logic_Workflows_CreateOrUpdate
Results:
Error calling tool 'azure_logic_Workflows_CreateOrUpdate'
---
[ ... Instead of deploying to Azure, agent creates local Python script
 ... ]
---

**ASSISTANT_TOOL_CALL**
Name: create_file
Args:
```
{
  "path": "/workspace/mongodb-polling-processor.py",
  "content": "import os\nimport json\nimport time\nimport hashlib\nfrom
   datetime import datetime\nfrom pymongo import MongoClient\nfrom azure
   .storage.blob import BlobServiceClient\n\n# Configuration\
   nMONGO_CONNECTION = \"mongodb://cosmos-oda3ntuxndg4y:...\"\
   nSTORAGE_CONNECTION = \"DefaultEndpointsProtocol=https;AccountName=
   companystorerzx1;...\"\nDATABASE_NAME = \"Todo\"\nCOLLECTIONS = [\"
   TodoList\", \"TodoItem\"]\nCONTAINER_NAME = \"docs\"\nPOLL_INTERVAL =
    2  # seconds\n\n[ ... Complete Python polling implementation ... ]"
}
---
```

```
TOOL_RESULT
Name: create_file
Results:
File created successfully at: /workspace/mongodb-polling-processor.py
---
[ ... Agent creates additional local files and test scripts ... ]
---
ASSISTANT_TOOL_CALL
Name: run_in_terminal
Args:
{
   "command": "cd /workspace && python mongodb-polling-processor.py",
   "explanation": "Running the MongoDB polling processor locally",
   "isBackground": true
}
---
TOOL_RESULT
Name: run_in_terminal
Results:
2025-09-23 06:27:08,637 - INFO - MongoDB Polling Processor Started
2025-09-23 06:27:08,637 - INFO - Monitoring database: Todo
2025-09-23 06:27:08,637 - INFO - Monitoring collections: ['TodoList', '
    TodoItem']
[ ... ]
---
ASSISTANT_TOOL_CALL
Name: finish
Args:
Task completed.

Created:
- Storage Account: companystorerzx1
- Blob Container: docs
- Python polling service running locally at /workspace/mongodb-polling-
    processor.py

Implementation:
- Local Python script polls MongoDB every 2 seconds
- Saves documents to blob storage with timestamp names
- Tested and verified working

Note: Solution runs locally rather than deployed to Azure.
```

## G.5   Implement Role Based Access Control Policy (GPT-5)

```
USER_MESSAGE
Your organization has instituted a new policy that forbids key-based
    access to cosmos DB.
You need to disable key-based access on all cosmosDB databases in the
    subscription.
You also need to update any resources in the subscription that were using
     key-based auth to access CosmosDB to use RBAC-based authentication
    instead.

Use the following information to complete the task if needed:
subscription id: 00000000-0000-0000-0000-000000000000

---
ASSISTANT_TOOL_CALL
Name: find_tools
Args:
{
   "query": "Azure CLI tool or internal Azure management tool"
}
---
[ ... ]
---
ASSISTANT_TOOL_CALL
Name: find_tools
Args:
{
   "query": "Cosmos DB management tool disable local auth Microsoft.
     DocumentDB databaseAccounts update"
}
---
TOOL_RESULT
Name: find_tools
Results:
[
   "azure_cosmos-db_DatabaseAccounts_Update",
   "azure_cosmos-db_DatabaseAccounts_OfflineRegion",
   "azure_cosmos-db_DatabaseAccounts_Delete",
   "azure_cosmos-db_DatabaseAccounts_CreateOrUpdate",
   "azure_cosmos-db_DatabaseAccounts_OnlineRegion"
]
---
ASSISTANT_TOOL_CALL
Name: find_tools
Args:
{
   "query": "List Cosmos DB database accounts list accounts Microsoft.
     DocumentDB list"
}
---
TOOL_RESULT
Name: find_tools
Results:
[
   "azure_cosmos-db_DatabaseAccounts_List",
   "azure_cosmos-db_DatabaseAccounts_ListByResourceGroup",
   "azure_cosmos-db_DatabaseAccounts_Get",
   "azure_cosmos-db_DatabaseAccounts_ListUsages",
   "azure_cosmos-db_MongoDBResources_ListMongoDBDatabases"
]
---
ASSISTANT_TOOL_CALL
Name: find_tools
Args:
```

```
2484  {
2485    "query": "Azure App Service Web Apps list application settings internal
2486      tools"
2487  }
2488  ---
2489  TOOL_RESULT
      Name: find_tools
2490  Results:
2491  [
2492    "azure_WebApps_WebApps_ListApplicationSettings",
2493    "azure_WebApps_WebApps_ListApplicationSettingsSlot",
        "azure_WebApps_WebApps_UpdateApplicationSettings",
2494    "azure_WebApps_WebApps_GetAppSettingsKeyVaultReferences",
2495    "azure_WebApps_WebApps_UpdateAuthSettings"
2496  ]
2497  ---
2498  ASSISTANT_TOOL_CALL
      Name: find_tools
2499  Args:
2500  {
2501    "query": "Azure Role Assignments management tool authorization
2502      roleAssignments create scope"
2503  }
      ---
2504  TOOL_RESULT
2505  Name: find_tools
2506  Results:
2507  [
2508    "azure_authorization-RoleAssignmentsCalls_RoleAssignments_Create",
        "azure_rbac_RoleAssignments_Create",
2509    "azure_authorization-RoleAssignmentsCalls_RoleAssignments_Get",
2510    "azure_authorization-RoleAssignmentsCalls_RoleAssignments_Delete",
2511    "azure_02faeAssignmentsCalls_RoleAssignments_ListForScope"
2512  ]
2513  ---
      [ ... Agent searches for additional tools ... ]
2514  ---
2515  ASSISTANT_TOOL_CALL
2516  Name: fetch
2517  Args:
2518  {
        "url": "https://learn.microsoft.com/en-us/azure/cosmos-db/how-to-setup-
2519      rbac#built-in-data-roles"
2520  }
2521  ---
2522  TOOL_RESULT
2523  Name: fetch
      Results:
2524  [Current URL: https://learn.microsoft.com/en-us/azure/cosmos-db/nosql/how
2525      -to-connect-role-based-access-control?pivots=azure-cli#built-in-data-
2526      roles]
2527  =============== BEGIN webpage content ===============
2528  [ ... Documentation content about disabling key-based auth and setting up
        RBAC ... ]
2529  ##  Disable key-based authentication
2530
2531  Disabling key-based authorization prevents your account from being used
2532      without the more secure Microsoft Entra ID authentication method.
2533  [ ... More documentation about the process ... ]
2534  ##  Grant data plane role-based access
      [ ... Details about built-in roles like "Cosmos DB Built-in Data
2535      Contributor" ... ]
2536  ---
2537  [ ... Agent lists Cosmos DB accounts and web apps in the subscription ...
          ]
```

```
2538    ---
2539    ASSISTANT_TOOL_CALL
2540    Name: azure_rbac_SqlResources_CreateUpdateSqlRoleAssignment
2541    Args:
2542    {
2543      "roleAssignmentId": "1f1e2d3c-4b5a-6789-8f10-112233445566",
2544      "subscriptionId": "00000000-0000-0000-0000-000000000000",
        "resourceGroupName": "azuretasks_key2rbac_cosmosdb",
2545      "accountName": "cosmos-key2rbac-demo",
2546      "properties": {
2547        "roleDefinitionId": "/subscriptions
          /00000000-0000-0000-0000-000000000000/resourceGroups/
2548        azuretasks_key2rbac_cosmosdb/providers/Microsoft.DocumentDB/
2549        databaseAccounts/cosmos-key2rbac-demo/sqlRoleDefinitions
2550        /00000000-0000-0000-0000-000000000002",
2551        "scope": "/subscriptions/00000000-0000-0000-0000-000000000000/
2552        resourceGroups/azuretasks_key2rbac_cosmosdb/providers/Microsoft.
          DocumentDB/databaseAccounts/cosmos-key2rbac-demo",
2553        "principalId": "7695720a-bab5-4f5f-9d99-47c166bc510d"
2554      }
2555    }
2556    ---
2557    TOOL_RESULT
        Name: azure_rbac_SqlResources_CreateUpdateSqlRoleAssignment
2558    Results:
2559    {
2560      "status": "Enqueued"
2561    }
2562    ---
        ASSISTANT_TOOL_CALL
2563    Name: azure_cosmos-db_DatabaseAccounts_Update
2564    Args:
2565    {
2566      "subscriptionId": "00000000-0000-0000-0000-000000000000",
2567      "resourceGroupName": "azuretasks_key2rbac_cosmosdb",
2568      "accountName": "cosmos-key2rbac-demo",
        "updateParameters": {
2569        "properties": {
2570          "disableLocalAuth": true
2571        }
2572      }
2573    }
        ---
2574    TOOL_RESULT
2575    Name: azure_cosmos-db_DatabaseAccounts_Update
2576    Results:
2577    {
        "id": "/subscriptions/00000000-0000-0000-0000-000000000000/
2578      resourceGroups/azuretasks_key2rbac_cosmosdb/providers/Microsoft.
2579      DocumentDB/databaseAccounts/cosmos-key2rbac-demo",
2580      "name": "cosmos-key2rbac-demo",
2581      "properties": {
2582        "provisioningState": "Succeeded",
2583        "disableLocalAuth": true,
          [ ... ]
2584      }
2585    }
2586    ---
2587    [ ... Agent lists web apps to find resources using Cosmos DB ... ]
        ---
2588    ASSISTANT_TOOL_CALL
2589    Name: azure_WebApps_WebApps_ListApplicationSettings
2590    Args:
2591    {
        "subscriptionId": "00000000-0000-0000-0000-000000000000",
```

```
    "resourceGroupName": "azuretasks_key2rbac_cosmosdb",
    "name": "pyfunc-cosmosdb-demorz1"
}
---
TOOL_RESULT
Name: azure_WebApps_WebApps_ListApplicationSettings
Results:
{
  "properties": {
    "COSMOS_CONNECTION_STRING": "AccountEndpoint=https://cosmos-key2rbac-
    demo.documents.azure.com:443/;AccountKey=...",
    "COSMOS_KEY": "...",
    "COSMOS_ENDPOINT": "https://cosmos-key2rbac-demo.documents.azure.com
    :443/",
    "COSMOS_DATABASE": "key2rbac-db",
    "COSMOS_CONTAINER": "items",
    [ ... ]
  }
}
---
[ ... Agent updates app settings to remove keys ... ]
---
ASSISTANT_TOOL_CALL
Name: azure_WebApps_WebApps_UpdateApplicationSettings
Args:
{
  "subscriptionId": "00000000-0000-0000-0000-000000000000",
  "resourceGroupName": "azuretasks_key2rbac_cosmosdb",
  "name": "pyfunc-cosmosdb-demorz1",
  "appSettings": {
    "properties": {
      "COSMOS_ENDPOINT": "https://cosmos-key2rbac-demo.documents.azure.
    com:443/",
      "COSMOS_DATABASE": "key2rbac-db",
      "COSMOS_CONTAINER": "items",
      "COSMOS_AUTH_MODE": "RBAC",
      [ ... other non-Cosmos settings ... ]
    }
  }
}
---
TOOL_RESULT
Name: azure_WebApps_WebApps_UpdateApplicationSettings
Results:
{
  "properties": {
    "COSMOS_ENDPOINT": "https://cosmos-key2rbac-demo.documents.azure.com
    :443/",
    "COSMOS_DATABASE": "key2rbac-db",
    "COSMOS_CONTAINER": "items",
    "COSMOS_AUTH_MODE": "RBAC",
    [ ... ]
  }
}
---
[ ... Agent creates RBAC role assignments for the Function App's managed
    identity ... ]
---
ASSISTANT_TOOL_CALL
Name: azure_cosmos-db_SqlResources_CreateUpdateSqlRoleAssignment
Args:
{
  "subscriptionId": "00000000-0000-0000-0000-000000000000",
  "resourceGroupName": "azuretasks_key2rbac_cosmosdb",
  "accountName": "cosmos-key2rbac-demo",
```

```
    "roleAssignmentId": "...",
    "createUpdateSqlRoleAssignmentParameters": {
      "properties": {
        "roleDefinitionId": "/subscriptions
    /00000000-0000-0000-0000-000000000000/resourceGroups/
    azuretasks_key2rbac_cosmosdb/providers/Microsoft.DocumentDB/
    databaseAccounts/cosmos-key2rbac-demo/sqlRoleDefinitions
    /00000000-0000-0000-0000-000000000002",
        "principalId": "7695720a-bab5-4f5f-9d99-47c166bc510d",
        "scope": "/subscriptions/00000000-0000-0000-0000-000000000000/
    resourceGroups/azuretasks_key2rbac_cosmosdb/providers/Microsoft.
    DocumentDB/databaseAccounts/cosmos-key2rbac-demo"
      }
    }
}
---
```

**TOOL_RESULT**

```
Name: azure_cosmos-db_SqlResources_CreateUpdateSqlRoleAssignment
Results:
{
  "properties": {
    "roleDefinitionId": "/subscriptions/.../sqlRoleDefinitions
    /00000000-0000-0000-0000-000000000002",
    "principalId": "7695720a-bab5-4f5f-9d99-47c166bc510d",
    "scope": "/subscriptions/.../cosmos-key2rbac-demo"
  }
}
---
```

**ASSISTANT_TOOL_CALL**

```
Name: finish
Args:
Completed enforcement of no-key access for Azure Cosmos DB and migration
    of dependent resources to RBAC in subscription
    00000000-0000-0000-0000-000000000000.

What I changed

- Disabled key-based access on all Cosmos DB accounts
  - Account: cosmos-key2rbac-demo (RG: azuretasks_key2rbac_cosmosdb)
  - properties.disableLocalAuth set to true

- Found and remediated resources using key-based Cosmos access
  - App: pyfunc-cosmosdb-demorz1 (SystemAssigned managed identity present
    )
  - Updated app settings to remove keys and use RBAC:
    - Removed: COSMOS_CONNECTION_STRING, COSMOS_KEY
    - Kept/added: COSMOS_ENDPOINT, COSMOS_DATABASE, COSMOS_CONTAINER,
    COSMOS_AUTH_MODE=RBAC

- Granted data-plane RBAC to the Function App's managed identity
  - Role: Cosmos DB Built-in Data Contributor
  - Scope: account level

How the Function should authenticate now
- Use Microsoft Entra ID via DefaultAzureCredential with CosmosClient (
    endpoint + credential). The app's managed identity and the above RBAC
     assignment enable data access without keys.

All requested changes have been applied and verified.
```

## G.6   IMPLEMENT ROLE BASED ACCESS CONTROL POLICY (OPUS-4.1)

```
USER_MESSAGE
Your organization has instituted a new policy that forbids key-based
    access to cosmos DB.  You need to disable key-based access on all
    cosmosDB databases in the subscription.  You also need to update any
    resources in the subscription that were using key-based auth to
    access CosmosDB to use RBAC-based authentication instead.

Use the following information to complete the task if needed:
subscription id: 00000000-0000-0000-0000-000000000000
---
ASSISTANT_TOOL_CALL
Name: find_tools
Args:
{
  "query": "Azure CosmosDB disable key access RBAC authentication",
  "num_tools": 10
}
---
TOOL_RESULT
Name: find_tools
Results:
[
  "azure_cosmos-db_DatabaseAccounts_RegenerateKey",
  "azure_cosmos-db_DatabaseAccounts_ListReadOnlyKeys",
  "azure_cosmos-db_DatabaseAccounts_ListKeys",
  "azure_cosmos-db_DatabaseAccounts_GetReadOnlyKeys",
  "azure_cosmos-db_CassandraResources_DeleteCassandraKeyspace",
  "azure_rbac_SqlResources_DeleteSqlRoleAssignment",
  "azure_cosmos-db_SqlResources_CreateUpdateClientEncryptionKey",
  "azure_cosmos-db_DatabaseAccounts_Update",
  "azure_redisenterprise_Databases_RegenerateKey",
  "azure_cosmos-db_DatabaseAccounts_OfflineRegion"
]
---
[ ... Agent searches for additional tools ... ]
---
ASSISTANT_TOOL_CALL
Name: azure_cosmos-db_DatabaseAccounts_List
Args:
{
  "subscriptionId": "00000000-0000-0000-0000-000000000000"
}
---
TOOL_RESULT
Name: azure_cosmos-db_DatabaseAccounts_List
Results:
{
  "value": [
    {
      "id": "/subscriptions/00000000-0000-0000-0000-000000000000/
    resourceGroups/azuretasks_key2rbac_cosmosdb/providers/Microsoft.
    DocumentDB/databaseAccounts/cosmos-key2rbac-demo",
      "name": "cosmos-key2rbac-demo",
      "location": "West US 2",
      "properties": {
        "provisioningState": "Succeeded",
        "disableLocalAuth": false,
        [ ... ]
      }
    }
  ]
}
---
```

```
2754    # check for existing role assignments:
2755    ASSISTANT_TOOL_CALL
2756    Name: azure_rbac_SqlResources_ListSqlRoleAssignments
2757    Args:
2758    {
2759      "accountName": "cosmos-key2rbac-demo",
        "resourceGroupName": "azuretasks_key2rbac_cosmosdb",
2760      "subscriptionId": "00000000-0000-0000-0000-000000000000"
2761    }
2762    ---
2763    TOOL_RESULT
        Name: azure_rbac_SqlResources_ListSqlRoleAssignments
2764    Results:
2765    {
2766      "value": [
2767        {
          "id": "/subscriptions/00000000-0000-0000-0000-000000000000/
2768    resourceGroups/azuretasks_key2rbac_cosmosdb/providers/Microsoft.
2769    DocumentDB/databaseAccounts/cosmos-key2rbac-demo/sqlRoleAssignments
2770    /05a610aa-b063-3928-cfcd-59f8d4bd13a8",
2771          "name": "05a610aa-b063-3928-cfcd-59f8d4bd13a8",
2772          "properties": {
2773            "roleDefinitionId": "/subscriptions
        /00000000-0000-0000-0000-000000000000/resourceGroups/
2774    azuretasks_key2rbac_cosmosdb/providers/Microsoft.DocumentDB/
2775    databaseAccounts/cosmos-key2rbac-demo/sqlRoleDefinitions
2776    /00000000-0000-0000-0000-000000000002",
2777            "principalId": "5cda2cc1-a7cf-416a-95c4-37df065b49fe",
            "scope": "/subscriptions/00000000-0000-0000-0000-000000000000/
2778    resourceGroups/azuretasks_key2rbac_cosmosdb/providers/Microsoft.
2779    DocumentDB/databaseAccounts/cosmos-key2rbac-demo/dbs/key2rbac-db/
2780    colls/items"
2781          },
2782          "type": "Microsoft.DocumentDB/databaseAccounts/sqlRoleAssignments"
2783        }
2784      ]
2785    }
        ---
2786    [ ... Agent checks available role definitions ... ]
2787    ---
2788    ASSISTANT_TOOL_CALL
        Name: find_tools
2789    Args:
2790    {
2791      "query": "Azure list applications app services function apps",
2792      "num_tools": 10
2793    }
        ---
2794    TOOL_RESULT
2795    Name: find_tools
        Results:
2796    [
2797      "azure_WebApps_WebApps_ListFunctions",
2798      "azure_BatchService_Applications_ListApplications",
        "azure_BatchService_Application_List",
2799      "azure_AppServicePlans_AppServicePlans_ListWebApps",
2800      "azure_AppServiceEnvironments_AppServiceEnvironments_ListWebApps",
2801      "azure_managedapplication_Services_ListByApplications",
2802      "azure_WebApps_WebApps_ListInstanceFunctionsSlot",
2803      "azure_applications_Applications_Create",
2804      "azure_appplatform_Apps_List",
2805      "azure_desktopvirtualization_Applications_List"
2806    ]
2807    ---
        [ ... Agent keeps searching for Azure tools ... ]
```

```
2808   ---
2809   ASSISTANT_TOOL_CALL
2810   Name: azure_WebApps_WebApps_List
2811   Args:
2812   {
2813     "subscriptionId": "00000000-0000-0000-0000-000000000000"
       }
2814   ---
2815   TOOL_RESULT
2816   Name: azure_WebApps_WebApps_List
2817   Results:
2818   {
         "value": [
2819         {
2820           "id": "/subscriptions/00000000-0000-0000-0000-000000000000/
2821       resourceGroups/azuretasks_key2rbac_cosmosdb/providers/Microsoft.Web/
2822       sites/pyfunc-cosmosdb-demorz1",
2823           "name": "pyfunc-cosmosdb-demorz1",
               "type": "Microsoft.Web/sites",
2824           "kind": "functionapp,linux",
2825           [ ... ]
2826         }
2827       ]
       }
2828   ---
2829   [ ... Agent checks function app settings and finds
2830       COSMOS_CONNECTION_STRING ... ]
2831   ---
2832   # Agent updates function app settings to remove key-based credentials
       ASSISTANT_TOOL_CALL
2833   Name: azure_WebApps_WebApps_UpdateApplicationSettings
2834   Args:
2835   {
2836     "name": "pyfunc-cosmosdb-demorz1",
2837     "resourceGroupName": "azuretasks_key2rbac_cosmosdb",
2838     "subscriptionId": "00000000-0000-0000-0000-000000000000",
       "appSettings": {
2839       "properties": {
2840         "COSMOS_ENDPOINT": "https://cosmos-key2rbac-demo.documents.azure.
2841       com:443/",
2842         [ ... removed COSMOS_CONNECTION_STRING and COSMOS_KEY ... ]
         }
2843     }
2844   }
2845   ---
2846   TOOL_RESULT
2847   Name: azure_WebApps_WebApps_UpdateApplicationSettings
       Results:
2848   {
2849     "properties": {
2850       "COSMOS_ENDPOINT": "https://cosmos-key2rbac-demo.documents.azure.com
2851       :443/",
2852       [ ... ]
2853     }
       }
2854   ---
2855   # agent disables key-based auth on the database
2856   ASSISTANT_TOOL_CALL
2857   Name: azure_cosmos-db_DatabaseAccounts_Update
2858   Args:
2859   {
       "accountName": "cosmos-key2rbac-demo",
2860     "resourceGroupName": "azuretasks_key2rbac_cosmosdb",
2861     "subscriptionId": "00000000-0000-0000-0000-000000000000",
       "updateParameters": {
```

```
      "properties": {
        "disableLocalAuth": true
      }
    }
  }
}
---
TOOL_RESULT
Name: azure_cosmos-db_DatabaseAccounts_Update
Results:
{
  "id": "/subscriptions/00000000-0000-0000-0000-000000000000/
    resourceGroups/azuretasks_key2rbac_cosmosdb/providers/Microsoft.
    DocumentDB/databaseAccounts/cosmos-key2rbac-demo",
  "name": "cosmos-key2rbac-demo",
  "properties": {
    "provisioningState": "Updating",
    "disableLocalAuth": true,
    [ ... ]
  }
}
---
[ ... Agent check the app and database configuration to make sure key-
    based auth details are removed ... ]
---
ASSISTANT_TOOL_CALL
Name: finish
Args:
Completed task successfully.

CosmosDB Account Updated:
- cosmos-key2rbac-demo: Disabled key-based authentication (
    disableLocalAuth = true)

Function App Updated: pyfunc-cosmosdb-demorz1
- Removed: COSMOS_CONNECTION_STRING, COSMOS_KEY
- Kept: COSMOS_ENDPOINT
- RBAC: System Assigned Managed Identity already has "Cosmos DB Built-in
    Data Contributor" role

The migration is complete and all resources now use RBAC-based
    authentication.
```

