# OpenReview forum: "TheMCPCompany: Creating General-purpose Agents with Task-specific Tools"
_ICLR.cc/2026/Conference — Submitted to ICLR 2026_

### Official Review · Reviewer_XMs6 · 2025-10-17

**Soundness:** 3
**Presentation:** 2
**Contribution:** 4
**Rating:** 8
**Confidence:** 4

**Summary:**

```
I used LLM to fix the grammar of the Official Review, but all opinions are my own
```
Many current AI agents rely heavily on general-purpose tools such as web browsers to perform tasks—whether retrieving data or operating software. This approach is inefficient and costly, and it fails to capture how most real-world professional work is done. For example, in enterprise environments that use Azure for cloud services or GitLab for code management, there exist specialized APIs or SDKs that are far more suitable for task automation, but current agents struggle to leverage them effectively.
To address this, the authors built a “tool library” that converts common enterprise services—like Azure, GitLab, and internal communication tools—into over 18,000 “AI-native tools” (called MCP tools). For instance, instead of manually updating database settings via the Azure UI, an AI agent can now directly call a dedicated “update Azure database version” tool.
They also constructed a simulated company environment (“TheMCPCompany”) to benchmark AI agents in realistic work settings. The environment includes both simple tasks (e.g., labeling files) and complex ones (e.g., fixing a faulty cloud service). A baseline agent (“MCPAgent”) is introduced, which must first discover relevant tools for a given task (e.g., finding “check Azure database version” or “restart service”) and then use them to solve the problem.

Although the paper doesn’t introduce a novel method, I find the problem setting very meaningful and the work potentially impactful. I recommend accepting this paper.

**Strengths:**

1. The idea of turning real-world software APIs into standardized, callable tools for AI agents is highly interesting.

2. The dataset and environment are both valuable community resources that can enable further research.

**Weaknesses:**

1. Some sections, especially the data construction process, are not clearly written.

2. It remains unclear how one might systematically improve the agent’s ability to use such tools efficiently.

**Questions:**

1. The paper seems to lack a clear explanation of how the data were constructed. The current narrative is somewhat scattered, and it’s hard to follow the full pipeline from tool creation to task setup.

2. This direction is quite open-ended, and I’m curious how one might improve generalization in this setting. Since your tools appear highly domain-specific, training on them might not help the model handle out-of-domain (OOD) scenarios. Have you considered splitting the tools and environments into disjoint train/test sets—for example, using part of the tools to fine-tune/RL a model like Qwen, and then testing on unseen tools or environments? Such an experiment would be very informative, and if you could include results along these lines, I’d be even more inclined to advocate for acceptance.

---

> ### Author Response · Authors · 2025-11-25
> **Response to reviewer XMs6**
>
> Thank you for your positive feedback and thoughtful comments. Below, we address your comments and questions and have also updated our paper to reflect these changes.
>
>
> > The paper seems to lack a clear explanation of how the data were constructed. The current narrative is somewhat scattered, and it’s hard to follow the full pipeline from tool creation to task setup.
>
> We have added Appendix D in our updated paper, which provides more details on how each task is modified and adapted to the MCP setting.
> Regarding the connection between tool creation and task setup, since our tool set exposes the full functionality of each service to the agent, it does not limit the actions available to the agent for each task. Therefore, unlike previous work that creates tasks based on the available tools, we treat tool creation and task setup as independent steps.
>
> >  I’m curious how one might improve generalization in this setting ... Have you considered splitting the tools and environments into disjoint train/test sets—for example, using part of the tools to fine-tune/RL a model like Qwen, …
>
> Thank you for the interesting suggestion.
> We have been thinking about this, and the main roadblock to fine-tuning models for our setup is the lack of large-scale training data.
> While sufficient for evaluation, the 192 tasks in TheMCPCompany are inadequate for large-scale fine-tuning.
> In our experience, creating high-quality data for complex environments is extremely difficult, and simple synthetic data generation methods are insufficient. In fact, we first attempted to generate our Azure tasks synthetically but failed to generate tasks that are challenging for frontier models. Instead, we relied on people with extensive experience with the Azure platform to create challenging Azure tasks, which is very time-consuming.
>
> While investigating potential methods for data generation and training is interesting, the main goal of our work is to establish the use of massive task-specific tool sets as a promising and effective approach for creating general-purpose agents.
> We encourage future work to build on top of TheMCPCompany’s extensive tool set and complex environment and explore novel methods for generating large-scale complex training tasks for agentic workflows.
>
>
> > This direction is quite open-ended / It remains unclear how one might systematically improve the agent’s ability to use such tools efficiently.
>
> We completely agree that there are many unanswered questions about creating general-purpose agents that perform well in practice, which highlights the value of our work in stimulating new research in this area. In addition to the data generation aspect mentioned above, we would like to discuss some of the directions that we have been mainly thinking about:
> - Developing better retrieval methods is one way to improve the agents' performance. In fact, our new experiments in Section 6 demonstrate the correlation between the retriever quality and agent performance.
> - Another possibility is to create a multi-agent system in which a small language model is dedicated to interacting with each service, and the main agent just orchestrates the service-specific agents. Since service-specific agents are smaller and only focus on one set of tools, they are easier to improve through fine-tuning. Moreover, as recent work advocates [1], this could further decrease the agents’ inference costs.
> - Our new analysis in Section 6 and Appendix A also shows that agents' systematic problem-solving skills degrade noticeably when dealing with more complex services such as Microsoft Azure. It would be very interesting to investigate the root causes of this phenomenon and explore potential solutions.
> - Finally, as mentioned in our limitations section, performance is not the only roadblock to deploying LLM agents in real-world applications. LLM agents can freely explore diverse actions with potentially harmful side effects (e.g., deleting a VM containing critical data or exposing sensitive information). In addition to performance, exploring other limitations of LLMs, such as these, would be very valuable in bringing LLM agents to real-world applications.
>
> [1] Small Language Models are the Future of Agentic AI (https://arxiv.org/abs/2506.02153)

---

> > ### Comment · Reviewer_XMs6 · 2025-11-25
> >
> > After reading your response and carefully going through the updated Appendix D, I unfortunately find that many of my original concerns remain unresolved. I had originally given this submission a fairly high score because I found the overall direction meaningful and the dataset potentially impactful. However, after examining the new details, I am not satisfied with the clarity, rigor, or completeness of the explanations you provided, and I will have to lower my score from 8 to 4.
> >
> > First, regarding the data construction process, the newly added Appendix D does not actually clarify the pipeline in a way that addresses my concerns. For example, the paper references a specific link for task descriptions, but that link is not accessible, so it is impossible for a reader to verify or even understand how the tasks were actually specified. More importantly, the description of “Ground Truth Tool Annotation” raises more questions than it answers. The text vaguely explains that tasks were decomposed into sub-tasks, and then each sub-task was mapped to a small set of required tools, but the process for annotating these sub-tasks is not made clear. There is no explanation of how correctness was verified, how many annotators were involved, what instructions they followed, how consistency was ensured, or how conflicts were resolved when combining sub-tasks into a full task. Given how crucial this annotation step is to the environment’s validity, a vague and high-level paragraph does not suffice.
> >
> > The explanation of the GitLab sub-task annotation using LLM-generated Python code is also under-specified. For example, it is unclear how you ensured that the generated scripts were indeed correct, how many scripts required manual correction, whether the LLM was prone to hallucinating APIs, how that was handled, whether the mapping from API endpoints to MCP tools was systematically audited, and so on. The description currently reads as if the process worked smoothly by default, but without any discussion of failure modes or verification procedures, it is difficult to have confidence in the quality of the annotations. Overall, the description feels imprecise and somewhat superficial, which is disappointing given the centrality of this component to the benchmark.
> >
> > Second, regarding my question about improving generalization and the possibility of creating disjoint train/test splits, I find your response surprisingly narrow and not sufficiently thoughtful. You mention that there are only 192 tasks and therefore not enough data for large-scale fine-tuning, and that generating high-quality tasks is difficult. But this does not actually answer the question of whether controlled generalization experiments are possible. There are now multiple agent tool-use datasets available publicly, and even within your existing environment, one could easily roll out multiple trajectories using strong models, create pass@k samples, and use those to produce additional data. With a mixture of rollout-generated trajectories and the existing tasks, one could at least attempt small-scale fine-tuning and then test on unseen tools or unseen services. This is a very straightforward idea that many recent agentic papers explore, so I find it surprising that the response treats this as something fundamentally blocked by the lack of data, rather than engaging with the actual experimental possibility.
> >
> > To summarize: I initially viewed this submission quite positively. However, the issues around data construction and ground-truth annotation remain unclear, and the responses reinforce my impression that the current version of the paper does not yet meet the level of rigor, transparency, and careful explanation that a benchmark paper should provide. Given these concerns and the lack of convincing clarification, I am lowering my score from 8 to 4. I still believe the direction is important, but the current presentation lacks the thoroughness necessary for confident acceptance.

---

> > > ### Author Response · Authors · 2025-11-27
> > >
> > > We thank the reviewer for their continued engagement.
> > > We understand the reviewer’s concerns regarding the rigor of our data construction process. We believe there are misunderstandings regarding the nature of the tool selection process (execution-based verification vs annotator consensus), which we clarify below. We have added additional details and examples in the paper to provide further clarification.
> > > ## Broken Link in the Task Description
> > >
> > > The link the reviewer found inaccessible is part of the original task description ([Full Task Description](https://github.com/TheAgentCompany/TheAgentCompany/blob/98b68ef82a47690c316f42fddb05baafaab56851/workspaces/tasks/sde-implement-hyperloglog/task.md?plain=1)) and is intentional. It points to the locally hosted GitLab instance in the agent’s evaluation environment and not a public website. We have added an extended task description, a clarifying note, and a screenshot from the local web page in Appendix D to resolve the confusion.
> > >
> > > ## Correctness of the Selected Tools
> > >
> > > We clarify that we have not used external annotators. The tools are selected by the authors and verified against the live environment.
> > >
> > > **Selection Process**: The authors are well familiar with the environment and for ownCloud, Plane, and RocketChat, have manually selected the tools required for each sub-task. The tools for GitLab sub-tasks are also selected by the authors, but with the help of LLMs (the authors still verified the results).
> > >
> > > **Correctness**: We did _not_ rely on our own judgment for assessing the correctness of the selected tools. Every set of tools was executed against the live environment to verify its correctness.
> > > For example, to verify the correctness of the selected tools for sending a RocketChat message, we called the tools against the environment and checked the RocketChat web interface to verify that the message was sent successfully.
> > >
> > > **Conflicts**: The reviewer asked about conflicts when merging the required tools for sub-tasks. The final tool set for each task is defined as the union of the required tools identified for its sub-tasks, without any potential for conflicts. To illustrate, we have added a new example in Appendix D.
> > >
> > > ## LLM-Generated Scripts and REST API to Tool Mapping
> > >
> > > The reviewer rightly pointed out the risk of LLM errors when generating the intermediate Python scripts. We want to clarify that **the LLM was used as a drafting aid, and the authors verified the correctness**.
> > >
> > > **Audit Process**: We treated all LLM-generated scripts as untrusted. We have executed every script against the live environment and verified the results using GitLab's web interface; e.g., for a script that closes a GitLab issue, we have executed the script and checked the web interface of GitLab to ensure the script successfully closed the issue.
> > >
> > > **Mistakes in Scripts**: If there are any mistakes in a generated script (e.g., hallucinated APIs), it would fail our audit. The authors then manually corrected any such failures until the script passed the audit.
> > >
> > > The reviewer also asked about the number of scripts that needed correction. Since such failures do not impact our final results (all scripts are manually verified and fixed if needed), we unfortunately did not keep track of the related statistics.
> > >
> > > **REST API to MCP Tool Mapping**: Since we have created our MCP tools from the REST APIs (Section 3.3), there is a deterministic one-to-one mapping between REST API endpoints and MCP tools, by design.
> > >
> > > ## Training Experiment and OOD Generalization
> > >
> > > We apologize for the miscommunication. We fully agree with the reviewer that such a training experiment is technically feasible. Moreover, we speculate that the model could potentially learn some universal problem-solving skills that transfer to unseen services and tool sets. Skills such as systematic troubleshooting and extended exploration (please see Section 6 and Appendix A, where we discuss such skills for Azure tasks).
> > >
> > > **Training data:**
> > > Except for our Azure tasks, most of the tasks from TheAgentCompany involve multiple services. It is possible to decompose the tasks into smaller tasks that depend only on a subset of services, as we did for Oracle tool selection, and create disjoint train/test sets. But these smaller tasks will be inherently simpler.
> > > As the reviewer mentioned, another option is to use existing tool-use datasets.
> > > Both of these approaches are valid and could potentially improve model performance on unseen services and tool sets.
> > >
> > > While this is an interesting experiment, we respectfully maintain that the primary focus of our work is to study the potential and limitations of general-purpose tool-based agents, and new training experiments shift the goalpost of our contribution.

---

> > > > ### Comment · Reviewer_XMs6 · 2025-11-28
> > > >
> > > > Thank you for the detailed follow-up. The additional clarifications on the task descriptions, the verification steps, and the audit process did clear up several of my earlier concerns. I appreciate the effort in expanding Appendix D and in explaining how the authors validated the correctness of tool selections and LLM-generated scripts.
> > > >
> > > > That said, one key issue remains for me. I still believe the paper would benefit greatly from **an explicit training / OOD generalization experiment**, even if small-scale. My reason is not to shift the goal of the paper, but because such an experiment would help characterize the nature of the proposed benchmark itself:
> > > >
> > > > 1. If a model fine-tuned on other MCP-style datasets achieves large gains on TheMCPCompany tasks, this would suggest the benchmark is relatively easy to transfer into and might be quickly saturated.
> > > > 2. If the gains are small, this would instead highlight the benchmark’s difficulty and further support its value as a realistic, challenging test bed.
> > > >
> > > > This type of analysis would make the contribution more complete. Even a lightweight version—such as fine-tuning a strong model on existing MCP tool-use datasets and evaluating on your benchmark—would already provide meaningful insight.
> > > >
> > > > Given the improvements in the revised explanation, I adjust my score to **6**. If the authors can provide a more thoughtful discussion or preliminary evidence regarding the OOD generalization question above—whether positive or negative—I would be happy to further raise my score to **8** to support acceptance.
> > > >
> > > > Thank you again for the careful revisions.

---

### Official Review · Reviewer_4jnk · 2025-10-30

**Soundness:** 4
**Presentation:** 3
**Contribution:** 3
**Rating:** 8
**Confidence:** 3

**Summary:**

This paper introduces TheMCPCompany, a benchmark for evaluating general-purpose AI agents that primarily interact with their environment through a large set of task-specific tools (over 18,000) rather than a few general-purpose tools like web browsers. The core contributions include: (1) the creation of this large-scale, realistic benchmark based on the Model Context Protocol (MCP), which includes complex tasks adapted from a software company simulator and new challenges for the Azure cloud platform; (2) an extensive evaluation comparing browser-based agents to agents with access to either a pre-selected 'oracle' tool set or a tool-retrieval mechanism; and (3) key findings that demonstrate the potential of task-specific tools to improve performance and reduce cost, while also exposing the significant difficulties agents face in navigating and combining thousands of tools in complex enterprise environments.

**Strengths:**

1. This paper tackles a very relevant and interesting angle: understanding the capabilities of general-purpose agents when they are equipped with large, heterogeneous tool collections. Studying how LLMs perform as the number of available tools scales is highly realistic and timely, given the fast-evolving ecosystem of MCP tools.

2. The writing and motivation are experienced and clear, making the paper's contributions easy to grasp. The design of the benchmark is intuitive and well-justified; it builds sensibly on prior work by replacing a few general-purpose tools with a massive set of task-specific ones, thereby creating a novel and challenging testbed.

3. The experiments yield meaningful and interesting observations. For instance, the note that GPT-5's excellent performance is partly due to its perseverance provides genuine insight that maps model behavior directly to success and failure patterns. The clear performance gap between models with oracle tools versus tool retrieval effectively pinpoints the current challenges in tool discovery and usage.

**Weaknesses:**

The main weakness lies in the insufficient details provided for the MCPAgent's tool-finding function. This module is central to the paper's investigation of agents in large-scale tool environments, yet its implementation is only briefly described. Specifically, the choice of the embedding model is a critical design decision that could significantly impact retrieval quality and, consequently, the overall agent performance. The authors state, "We use OpenAI’s text-embedding-3-large model," but there is no discussion or ablation study on how this choice affects the results. Would a different embedding model change the performance gap between models, especially for smaller ones like GPT-5-mini? Without this analysis, it's difficult to fully assess the robustness of the retrieval approach and the conclusions drawn from it.

**Questions:**

1. Could you provide more details on the tool finding function? For example, what was the value of k (the number of tools returned per query), and how was it determined, and were there any strategies for handling the diversity of tool schemas (e.g., name vs. description weighting) during embedding?

2. How sensitive are your key results, especially the poor performance of smaller models with retrieval, to the choice of the embedding model? Did you experiment with any other models, and if so, were the conclusions consistent?

3. The error analysis is insightful but brief. For the complex Azure tasks where all models failed, could you provide more detail on the specific types of reasoning failures? For example, were the issues more related to flawed problem decomposition, an inability to understand tool dependencies, or something else? A more detailed breakdown here would be very valuable for the community.

---

> ### Author Response · Authors · 2025-11-25
> **Response to reviewer 4jnk**
>
> Thank you for your positive feedback and detailed comments. Below, we address your comments and questions and have also updated our paper to reflect these changes.
>
> > Could you provide more details on the tool finding function? For example, what was the value of k (the number of tools returned per query), and how was it determined, and were there any strategies for handling the diversity of tool schemas (e.g., name vs. description weighting) during embedding?
>
> We have added Appendix E.1, which provides more details about the tool-finder function. Here is the short answer to the reviewer’s question:
> - By default, the tool-finder function returns five tools per query. However, it accepts a `num_tools` argument that the agent can use to change the number of tools returned for each query. We noticed that LLMs rarely set this argument when using the tool-finder function.
> - In our experiments, we first prefix the name of each tool with the name of the corresponding service (e.g., all GitLab tools are named “gitlab_TOOL_NAME”), which allows the agent and the retriever to associate tools with their corresponding services.  We then encode the entire JSON tool specification returned by the MCP servers without any additional changes (see Appendix E.1 for an example). Even with this simple approach, MCPAgent can effectively discover the required tools for each task. Therefore, we leave more complex embedding and retrieval strategies to future work.
>
> > How sensitive are your key results, especially the poor performance of smaller models with retrieval, to the choice of the embedding model? Did you experiment with any other models, and if so, were the conclusions consistent?
>
> Thank you for the interesting question. We have conducted new experiments with GPT-5-mini and GPT-4.1 using two top embedding models from the MTEB leaderboard: Qwen3-Embedding 0.6b and Qwen3-Embedding 4b (Section 6 and Appendix C in our updated paper).
>
> | | OpenAI Text Emb. | Qwen3 0.6B | Qwen3 4B |
> |---| --- | --- | --- |
> |GPT-4.1  | 37.10 | 39.54 | 41.44 |
> |GPT-5-mini  | 32.11 | 31.27 | 30.46  |
>
> We find that GPT-4.1’s performance improves with better embedding models. However, GPT-5-mini cannot leverage the better embedding model, and its performance even decreases with the stronger Qwen3-Embedding 4b model. Coupled with the fact that, also in our main experiments (Table 2), GPT-5-mini was the only model that completely failed to use tool retrieval, these results further strengthen our claim that effectively using massive tool sets to solve challenging problems in complex environments requires both better reasoning and retrieval models, and that one is not a substitute for the other.
>
> > The error analysis is insightful but brief. For the complex Azure tasks where all models failed, could you provide more detail on the specific types of reasoning failures? For example, were the issues more related to flawed problem decomposition, an inability to understand tool dependencies, or something else? A more detailed breakdown here would be very valuable for the community.
>
> Please see our general response to all reviewers as well as Section 6 and Appendix A in our updated paper.

---

### Official Review · Reviewer_bmgw · 2025-10-31

**Soundness:** 2
**Presentation:** 1
**Contribution:** 1
**Rating:** 2
**Confidence:** 3

**Summary:**

This paper provide a new benchmark to evaluate the tool use and tool calling of LLM agent with MCP tools. This benchmark is notable for its scale, including over 18,000 MCP tools.

**Strengths:**

1. The MCPAgent incorporates 18,000 tools and introduces a gateway MCP server to retrieve the tools relevant to each user query, thereby improving performance and reducing operational costs.
2. This paper evaluates the MCPAgent on challenging tasks that reflect the complexity of real scenarios.

**Weaknesses:**

1. Although constructing a standardized set of MCP tools requires substantial engineering effort, the novelty of this paper appears to be limited.
2. Some experiment setups are confusing. For example, in Table 2, the comparison between the **MCPAgent** and the **Oracle Tool Set** supports the claimed advantages of introducing a gateway MCP server. However, it is unclear why the **MCPAgent** is also compared with the **browser-based agent**, given that their functionalities and supported capabilities differ significantly.

**Questions:**

1. Could the authors further clarify the novelty of this work?
2. What is the reasoning behind comparing the **MCPAgent** with the **browser-based agent**? Their functionalities and supported capabilities differ substantially.
3. Is the MCP tool set fixed at 18,000 tools in the experiments? Does the benchmark support a dynamic tool set? A dynamic setting might better capture real-world scenarios where available tools evolve over time.

---

> ### Author Response · Authors · 2025-11-25
> **Response to reviewer bmgw**
>
> Thank you for your thoughtful comments on our paper. Below, we address your comments and questions.
>
> > Could the authors further clarify the novelty of this work?
>
> Our work establishes tool-based agents as a cheaper and more effective alternative to browser-based agents in complex enterprise environments. To provide solid and reliable findings, we have also made the following contributions to the community:
>
> - We have created TheMCPCompany, a benchmark that provides more than 18,000 functional MCP tools for interacting with complex services like Azure, enabling evaluation in real-world environments.
> - We have created MCPAgent as a baseline for tool-based agents, allowing us to directly compare tool-based and browser-based agents and study the challenges and opportunities of using tool-based agents in practice.
> - We have conducted extensive experiments to demonstrate the benefits of tool-based agents in comparison to browser-based agents (54% cheaper and 5.39 points better performance). Also using our ground-truth tool sets, we demonstrate the potential of tool-based agents for yielding even bigger improvements in performance with better tool retrieval methods (up to 13.79 points), motivating future work in this direction.
>
> We also quote other reviewers regarding the impact of our work. Quote from reviewer "4jnk": _"This paper tackles a very relevant and interesting angle …. Studying how LLMs perform as the number of available tools scales is highly realistic and timely, given the fast-evolving ecosystem of MCP tools."_ And, quote from reviewer "XMs6": _"I find the problem setting very meaningful and the work potentially impactful."_
>
> > What is the reasoning behind comparing the MCPAgent with the browser-based agent? Their functionalities and supported capabilities differ substantially.
>
> Our comprehensive tool set exposes the full functionality of the services used in our benchmark (e.g., GitLab). Therefore, MCPAgent, coupled with our comprehensive tool set, provides a direct alternative to web-browsing agents that rely on web interfaces to interact with the environment.
> This comparison is central to understanding the benefits of tool-based agents (such as MCPAgent) relative to the alternative browser-based agents. Using this comparison, our results show that tool-based agents are a cheaper, faster, and more effective way to create general-purpose agents.
>
> > Is the MCP tool set fixed at 18,000 tools in the experiments? Does the benchmark support a dynamic tool set? A dynamic setting might better capture real-world scenarios where available tools evolve over time.
>
> To compare results, we use the same tool set across all our experiments. However, our pipeline supports adding new services and MCP tools to the environment. Adding new tools is as easy as adding the URL or shell command of the new MCP server to a config file. We will release our code and data after the double blind review process, allowing the community to explore MCP servers with different tool sets.

---

> > ### Comment · Reviewer_bmgw · 2025-11-26
> >
> > Thank the authors for their detailed response. I have carefully reviewed the revised manuscript. I acknowledge the significant workload involved in normalizing 18,000 tools for MCP and appreciate the contribution of open-sourcing this platform to the community. However, as a benchmark study, the current insights derived from the experimental results remain somewhat limited. I encourage the authors to explore further comprehensive analyses to uncover more novel and significant findings.

---

### Official Review · Reviewer_uqnP · 2025-10-31

**Soundness:** 2
**Presentation:** 3
**Contribution:** 2
**Rating:** 2
**Confidence:** 4

**Summary:**

This paper presents TheMCPCompany, a benchmark and evaluation framework for testing LLM-based agents equipped with Model Context Protocol (MCP) tools. The work extends TheAgentCompany by integrating various enterprise services (Azure, GitLab, RocketChat, Plane, ownCloud) through the MCP interface, resulting in over 18,000 tool endpoints. The authors also propose MCPAgent, a tool-retrieval agent capable of discovering and invoking MCP tools automatically. Experiments are conducted on several proprietary models (GPT-4.1, GPT-5, Claude Sonnet, Opus), showing that MCP-based agents outperform browser-based baselines in cost and accuracy. The paper aims to highlight the potential of large-scale MCP environments for real-world agent evaluation.

**Strengths:**

- Strong engineering contribution: Implements a large, fully functional MCP benchmark with 18,000+ tools across enterprise services.
- Systematic evaluation pipeline: Builds upon TheAgentCompany with added realism (Azure integration).
- Empirical comparison: Includes quantitative cost and accuracy analysis between MCP and browser-based setups.
- Reproducibility commitment: The authors intend to release code, MCP servers, and Terraform scripts.

**Weaknesses:**

- Limited model coverage: Only closed-source models from OpenAI and Anthropic are evaluated; Gemini and other open-source models (eg., DeepSeek-V3, Qwen3, Llama) are excluded. This limits generalizability.
- Lack of retrieval comparison: The paper does not directly compare MCPAgent with traditional retrieval-based methods, making it unclear whether MCPAgent offers genuine advantages.
- Narrow task scope: The actual benchmark tasks are mainly Azure tasks, and other major components (e.g., TheAgentCompany) are reused without meaningful extension. As a benchmark paper, this is insufficient.
- Weak analysis of MCPAgent: The paper provides little insight into how MCPAgent performs tool discovery or why it succeeds/fails in specific cases.
- 18,000-tool claim not substantiated: Although the paper highlights a huge MCP tool set, it never reports how many tools are actually useful or invoked during evaluation.
- Lack of concrete examples: The two main contributions—Azure tasks and MCPAgent—are not illustrated with examples or reasoning traces, making the work difficult to interpret and assess.
- Lack of comparison with existing MCP benchmarks: The paper does not include direct comparisons with other MCP-based benchmarks (eg., MCPVerse or LiveMCPBench).

**Questions:**

- How many of the 18,000 MCP tools are actually used in the benchmark? Can you provide tool invocation statistics?
- Why were open-source models (e.g., DeepSeek-V3, Qwen3, and Llama) excluded from the evaluation?
- Can you show a concrete example of an Azure composite task and how MCPAgent solves (or fails to solve) it?
- How does MCPAgent compare to traditional retrieval-based systems or other MCP-agent implementations?
- What are the main factors that limit agent performance on Azure composite tasks?

---

> ### Author Response · Authors · 2025-11-25
> **Response to reviewer uqnP (1/3)**
>
> Thank you for your meticulous and thoughtful comments on our paper. Below, we address your comments and concerns and have also updated our paper to reflect these changes.
>
> First, we would like to clarify that our work goes beyond just evaluating tool calling with LLMs. Our main goal is to explore an entirely new paradigm for creating _general-purpose agents_. Specifically, our main experiments and artifacts are designed to answer two main questions:
> - Is it possible to create general-purpose agents that use massive task-specific tool sets to interact with complex environments?
> - What are the benefits of such agents compared to the alternative browser-based agents?
>
> And, our main finding is that tool-based agents are not only practical but also cheaper and more effective than browser-based agents.
>
> > Why were open-source models (e.g., DeepSeek-V3, Qwen3, and Llama) excluded from the evaluation?
>
> As mentioned above, our goal is to explore whether using task-specific tools is a viable paradigm for creating general-purpose agents, which requires pushing the scale of the tool set and the complexity of the environment to extremes. Therefore, we use the best frontier models to minimize the impact of current model limitations and study the potential of tool-based agents as a new design paradigm. Using these models, our work establishes tool-based agents with massive task-specific tool sets as an affordable, fast, and effective approach for creating general-purpose agents in the future as the number of available MCP tools increases.
>
>
> It is likely that open-source models struggle to use massive tool sets through retrieval. In fact, in our experiments, even smaller proprietary models (e.g., GPT-5-mini in Table 2) cannot effectively use task-specific tools through tool retrieval. However, our results demonstrate the significant potential of tool-based agents and encourage future work to investigate the limitations and further improve the capabilities of current models in using massive tool sets. Moreover, we believe TheMCPCompany facilitates future work by providing a massive tool set and a complex environment for such investigations.
>
>
> > The paper does not directly compare MCPAgent with traditional retrieval-based methods, making it unclear whether MCPAgent offers genuine advantages.
>
> We have added new experiments (Section 6 and Appendix C) where we compare MCPAgent’s tool retrieval performance with standard tool retrieval. For standard tool retrieval, we use the task description as the query and retrieve the top-k most similar tools based on the cosine similarity.
>
>
> | Method      | # Retrieved Tools | Recall |
> | --- | --- | --- |
> | MCPAgent (Sonnet 4)   | 15.7 | 60.0 |
> | MCPAgent (Opus 4.1)   | 25.8 | 69.7 |
> | MCPAgent (GPT-4.1)   | 13.5| 44.9 |
> | MCPAgent (o3)   | 22.2| 53.1 |
> | MCPAgent (GPT-5-mini)   | 20.2 | 32.8 |
> | MCPAgent (GPT-5)   | 15.3 | 58.7 |
> | Standard Retrieval | 10 | 11.7 |
> | Standard Retrieval| 15 | 14.7 |
> | Standard Retrieval | 20 | 16.2 |
> | Standard Retrieval| 25 | 17.9 |
>
>
> Even with the least effective model (GPT-5-mini), MCPAgent significantly outperforms standard tool retrieval.
> These results show the effectiveness of MCPAgent in exploring massive tool sets and discovering the tools needed at each step to complete complex tasks. Moreover, these results also support our claim that simple dense retrieval is not sufficient for complex real-world tasks in which task descriptions do not mirror the names and descriptions of the required tools.

---

> > ### Author Response · Authors · 2025-11-25
> > **Response to reviewer uqnP (2/3)**
> >
> > > Narrow task scope: The actual benchmark tasks are mainly Azure tasks, and other major components (e.g., TheAgentCompany) are reused without meaningful extension. As a benchmark paper, this is insufficient.
> >
> >
> > While we build on top of TheAgentCompany, our benchmark provides a fundamentally new capability and enables us to study a new category of agents that use massive task-specific tool sets to complete diverse tasks, which is not possible with TheAgentCompany.
> >
> > To make this possible, in addition to creating a massive and functional MCP tool set, we have also modified the original TheAgentCompany tasks for the MCP setting. These modifications are briefly mentioned in section 3.3 of our original paper. And, we have added Appendix D in the updated paper, which provides more details on the tool annotation and task modification processes.
> > Here, we summarize all the differences that distinguish our new benchmark, TheMCPCompany, from TheAgentCompany:
> >
> > - We provide a massive tool set (18,000+ tools) that exposes the full functionality of complex real-world services like Azure to the agent, enabling the evaluation of tool-based agents in a more realistic environment.
> > - We modify the original task descriptions, which are written for browser agents, to be compatible with tool-based agents.
> > - We update the evaluation script for each task to be compatible with both browser-based and tool-based agents.
> > - We annotate a small set of oracle tools for each task, which are sufficient for its successful completion. The oracle tool sets allow us to isolate the impact of tool discovery and study the upper bound on the performance of tool-based agents.
> > - We provide a set of Azure tasks that measure LLMs’ ability to complete complex enterprise tasks in a complex environment. In addition to descriptions, Azure tasks include an evaluation script and a setup and teardown script (using Terraform) to ensure reproducible results.
> >
> > Moreover, we consider building on top of TheAgentCompany a strength of our work, as it allows us to directly compare browser-based and tool-based agents and provide reliable conclusions without confounding factors.
> >
> > Finally, we quote reviewer "4jnk" regarding our choice to build on top of TheAgentCompany: _"The design of the benchmark is intuitive and well-justified; it builds sensibly on prior work by replacing a few general-purpose tools with a massive set of task-specific ones, thereby creating a novel and challenging testbed."_
> >
> >
> > > Weak analysis of MCPAgent: The paper provides little insight into how MCPAgent performs tool discovery
> >
> > We have conducted a new analysis (Section 6 and Figure 3) that provides insights into tool-discovery patterns used by MCPAgent. At a high level, MCPAgent manages to find the necessary tools by generating a series of search queries that are directly related to the required tools and their dependencies. Here, we summarize the main findings:
> >
> > - MCPAgent breaks each task into smaller subtasks and generates separate queries for each subtask, enabling targeted and more accurate search results.
> > - Stronger models, like GPT-5, frequently infer inter-tool dependencies from the arguments of retrieved tools and issue new search queries to find them. For example, if the agent is asked to send a message to “John Doe,” but it finds a tool that requires the person’s username (not their first and last name), the agent realizes that it needs additional tools and generates a new search query to find tools that can obtain the username from the first and last name.
> > - However, smaller models are not usually proactive in identifying tool dependencies. Instead, they often try the first tool with the wrong arguments and, after receiving an error from the environment, search for additional tools that are required.
> > - Finally, stronger models, especially GPT-5, do not give up easily; if they fail to find the correct tool on the first attempt, they update their search query and keep searching until they retrieve the required tools.
> >
> > Please see section 6 in our updated paper for examples and additional details.

---

> ### Author Response · Authors · 2025-11-25
> **Response to reviewer uqnP (3/3)**
>
> > why it [MCPAgent] … fails in specific cases.
>
>
> To provide insights into the new challenges introduced by tool retrieval, we have conducted a new analysis where we inspect trajectories for tasks where GPT-4.1 fails with tool retrieval but succeeds with the oracle tool set. We found that search errors accounted for a smaller proportion of task failures than one might expect.
> Instead, the addition of tool retrieval degrades the model's instruction following capabilities in multi-step tasks, which is the most common cause for failure, with two recurring patterns
>
> - If the agent first discovers tools that are similar to the ground truth tools but offer slightly different functionality, it uses them to implement a solution that appears correct but differs from the task instructions. For example, when the agent is asked to send a message to a specific user, if the agent first finds the tool for sending a message to a channel, it will post the message to the channel and tag the user, which is not the correct solution.
> - In 50% of the cases, the agent even finds the correct tools but fails to follow the precise instructions given in the task. For example, it does not follow the specified formatting rules when creating a spreadsheet. Note that the same model with access to the oracle tool set successfully completes these tasks. This suggests that the increased context length and cognitive load from tool retrieval degrade the model's instruction-following capabilities.
>
> Please see Section 6 and Appendix B in our updated draft for exact results and more details.
>
> > What are the main factors that limit agent performance on Azure composite tasks?
>
> Please see our general response to all reviewers as well as Section 6 and Appendix A in our updated paper.
>
> > Lack of concrete examples: The two main contributions—Azure tasks and MCPAgent—are not illustrated with examples or reasoning traces, making the work difficult to interpret and assess.
>
> We have updated our paper to provide examples of MCPAgent trajectories for several complex Azure tasks. Please see Appendix A for the discussion of the tasks and model behavior, and Appendix G for the exact agent trajectories (because the trajectories are very long, we do not include them here on OpenReview).
>
> > 18,000-tool claim not substantiated: Although the paper highlights a huge MCP tool set, it never reports how many tools are actually useful or invoked during evaluation.
>
> Tool invocation statistics are reported in Tables 4 and 5 for TheAgentCompany and Azure tasks, respectively.
>
> Based on our oracle tool sets, on average, 6.5 tools are required per task.
> We would like to highlight the distinction between the search space (18,000+ tools) and the solution path (average 6.5 tools).
> Finding the correct tools for complex tasks in a massive search space is one of the main challenges of tool-based agents in enterprise environments, and reducing the number of tools would artificially simplify the retrieval challenge.
>
> Moreover, using a massive search space that includes hard-to-detect distractor tools is crucial for understanding the challenges that agents face in the real world. For example, our new analysis (in Section 6) shows that using a tool similar to the ground truth tool but with slightly different functionality accounts for many of GPT-4.1's failures. Please see our response to your question about MCPAgent failures for more details on this failure mode.
>
>
> > Lack of comparison with existing MCP benchmarks: The paper does not include direct comparisons with other MCP-based benchmarks (eg., MCPVerse or LiveMCPBench).
>
> Thank you for mentioning these interesting works. We have already cited MCPVerse and LiveMCPBench in the related work section and noted that their main shortcoming is their small tool set.
>
> MCPVerse and LiveMCPBench include 550 and 527 tools, respectively, which are much fewer than the number of tools a general-purpose agent is exposed to in real-world applications. On the other hand, we provide 18,000+ tools that expose the full functionality of complex services such as Azure, which allows us to study the challenges of general-purpose tool-based agents in a realistic environment.
>
> Moreover, while MCPVerse and LiveMCPBench mainly focus on LLMs' tool-calling capabilities at a small scale, we explore if and how massive MCP tool sets can be used to create general-purpose _agents_. Towards this end, our experiments go beyond just evaluating tool calling capabilities and compare tool-based agents with browser-based agents and show that tool-based agents are a cheaper, faster, and more effective approach for creating general-purpose agents.

---

### Author Response · Authors · 2025-11-25
**To all reviewers**

We thank the reviewers for taking the time to review our paper and share their feedback. Below, we list the changes to our paper and then address the comment regarding additional analysis of our composite Azure tasks, shared by multiple reviewers.

In response to the reviewers' comments, we have extended our paper and added the following sections:
- Additional Analysis (Section 6)
- Detailed Analysis of Azure Composite Tasks (Appendix A)
- MCPAgent Error analysis on Adapted Tasks from TheAgentCompany (Appendix B)
- Tool Retrieval Analysis (Appendix C)
- Task Adaptation and Tool Annotation Details (Appendix D)
- Additional Details on Tool Finding Function Implementation (Appendix E.1)
- Examples of MCPAgent Trajectories for Complex Azure Tasks (Appendix G)

## Azure Composite Task Analysis

By integrating Azure into the environment, TheMCPCompany allows us to study agents’ behavior in complex real-world applications. In our experiments, we found that even the best LLMs struggle with complex tasks in complex environments like Azure. As suggested by reviewers uqnP and 4jnk, we have conducted a new analysis that provides insights into causes of this failure (Section 6, Appendix A, and Appendix G in our updated paper). This analysis also serves as an example for the type of insights that a realistic environment, like TheMCPCompany, provides into agentic workflows.

In our new analysis, we find that the systematic problem-solving skills of even the best LLMs diminish significantly in complex environments like Azure. Here, we discuss some recurring failure patterns on our composite Azure tasks.

- **No systematic troubleshooting**: Given a complex problem, models make specific assumptions about the root causes without any troubleshooting. For example, when a web app cannot access the database, GPT-5 assumes that incorrect database secrets and access management issues are causing the problem, without any evidence.
- **Incomplete Solutions**: For complex, multi-step Azure tasks, models often only implement part of the solution. For example, when asked to implement a new access policy, the agent updates the Azure web app settings to assign the required roles but does not update the application’s source code to use the new authentication method.
- **Limited Exploration**: For composite tasks in the Azure environment, models do not explore alternative solutions when their initial attempts to solve the problem fail. For instance, if the agent fails to deploy a specific resource due to insufficient quota, it does not try deploying it in a different region or using other resource types with similar functionality.
- **Little to no Testing**: Agents often do not check if the implemented solution was successful. For instance, after implementing a new access policy, the agent does not test if the application still functions as expected with the new changes.


Importantly, this is in contrast to models’ behavior in the simpler TheAgentCompany environment, where the best models systematically diagnose the issue, try multiple solutions, and test their final answer for correctness (please see our updated draft for more details and examples).
We believe this analysis just scratches the surface, and complex real-world environments introduce many other challenges that are not often present in simpler environments commonly used for evaluation. We encourage future work to use TheMCPCompany’s realistic environment to further explore the limitations and improve the capabilities of current models and agentic designs for real-world use-cases.

---

### Comment · Reviewer_XMs6 · 2025-11-28

Hi AC, I would like to adjust my score for this submission, but it seems that the system no longer allows me to modify the rating. Could you please check if this is a system restriction or if scoring has already been locked? Thank you!

---

> ### Comment · Area_Chair_m3dD · 2025-11-28
>
> Dear PCs and SACs,
>
>  Can you help to check if the problem exists?

---

> ### Comment · Area_Chair_m3dD · 2025-11-28
>
> Dear reviewer,
>
>   You can directly comment here for the score updating if the problem is still not solved later. I will take it into account.

---

### Meta-Review · Area_Chair_HqYV · 2026-01-01

**Summary:**

Reviewers agreed that the paper represents a substantial engineering effort and introduces a large-scale MCP-based environment. However, significant concerns were raised regarding benchmark rigor, novelty, and clarity of methodology. In particular, multiple reviewers questioned whether the work goes beyond infrastructure construction to deliver clear, generalizable scientific insights, as expected for an ICLR benchmark paper. Concerns about unclear data construction and annotation procedures, limited analytical depth, and over-reliance on proprietary models weighed heavily against acceptance.

**Reviewer Concerns:**

The rebuttal addressed several surface-level issues by adding appendices, examples, and additional analyses. However, core concerns remain unresolved, especially regarding the transparency and rigor of task construction and ground-truth annotation, as well as the lack of controlled generalization or training/OOD experiments. While the authors clarified execution-based verification, reviewers remained unconvinced that the benchmark design and analysis meet the level of rigor expected for a flagship conference contribution.

**Reviewer Scores:**

Reviewer uqnP: Likely unchanged (remains reject), as concerns about scope, analysis depth, and benchmark validity persist.

Reviewer bmgw: Unchanged (reject), maintaining the view that novelty and insight are limited despite engineering scale.

Reviewer 4jnk: Likely unchanged (positive), already supportive and less concerned with benchmark rigor.

Reviewer XMs6: Already lowered score during discussion; unlikely to increase further without new experiments.

---

### Decision · Program_Chairs · 2026-01-26

Reject